# Fat-associated lymphoid clusters control local IgM secretion during pleural infection and lung inflammation

Lucy H. Jackson-Jones[1,2], Sheelagh M. Duncan[1], Marlène S. Magalhaes[2], Sharon M. Campbell[1], Rick M. Maizels[1], Henry J. McSorley[1,3], Judith E. Allen[1,4,*] & Cécile Bénézech[2,*]

Fat-associated lymphoid clusters (FALC) are inducible structures that support rapid innate-like B-cell immune responses in the serous cavities. Little is known about the physiological cues that activate FALCs in the pleural cavity and more generally the mechanisms controlling B-cell activation in FALCs. Here we show, using separate models of pleural nematode infection with *Litomosoides sigmodontis* and *Altenaria alternata* induced acute lung inflammation, that inflammation of the pleural cavity rapidly activates mediastinal and pericardial FALCs. IL-33 produced by FALC stroma is crucial for pleural B1-cell activation and local IgM secretion. However, B1 cells are not the direct target of IL-33, which instead requires IL-5 for activation. Moreover, lung inflammation leads to increased IL-5 production by type 2 cytokine-producing innate lymphoid cells (ILC2) in the FALC. These findings reveal a link between inflammation, IL-33 release by FALC stromal cells, ILC2 activation and pleural B-cell activation in FALCs, resulting in local and antigen-specific IgM production.

[1] Institute of Immunology and Infection Research, University of Edinburgh, Edinburgh, Scotland EH9 3FL, UK. [2] Centre for Cardiovascular Science, University of Edinburgh, Edinburgh, Scotland EH16 4TJ, UK. [3] Centre for Inflammation Research, University of Edinburgh, Edinburgh, Scotland EH16 4TJ, UK. [4] Faculty of Biology, Medicine & Health, University of Manchester, AV Hill Building, Manchester M13 9PT, UK. * These Authors contributed equally to this work. Correspondence and requests for materials should be addressed to J.E.A. (email: judi.allen@manchester.ac.uk) or to C.B. (email: cbenezec@exseed.ac.uk).

The serous membranes covering the viscera and the wall of the body cavities define three fluid-filled cavities: the peritoneal, pleural and pericardial cavities. These serous cavities constitute important reservoirs of innate-like B-cell subsets, also called B1 cells, the major innate function of which is to ensure early immune protection from infection by rapid secretion of natural IgM. How and where natural IgM are secreted is not fully understood. Natural IgM antibodies do not undergo affinity maturation and thus bind antigens with overall low affinity. Although pentameric structures highly increase the avidity of IgM[1], such arrangements also limit diffusion into tissues, meaning that secretion into the circulation does not guarantee efficacy at the site of infection. Paradoxically, many studies have reported that peritoneal cavity B1 cells do not secrete antibodies either at steady state or upon peritoneal cavity challenge[2–5]. Upon activation, peritoneal B1 cells can relocate to the red pulp of the spleen, where they start producing IgM enabling secretion into the circulation[4,6–9], or to the intestine for secretion of IgM and IgA at the mucosal barrier[9–11]. Immune protection of the peritoneal cavity is orchestrated by inducible lymphoid structures found within certain visceral adipose tissue deposits: the milky spots of the omentum and fat-associated lymphoid clusters (FALC) of the mesenteries[9,12–15]. Upon immune challenge, these structures support rapid activation of serous B cells and germinal center formation[13,15]. The existence of similar lymphoid structures has been reported in the adipose deposits of the pleural cavity, the mediastinum[13,16–18] and the pericardium[13]. Although the density of FALCs in pericardium and mediastinum is high[13], the functional role of these clusters has not been investigated. Critically, the pleural cavity is an immune site of medical importance for the understanding of airway associated diseases[19], but little is known about the role of pleural B cells or the mechanisms controlling their function.

In an earlier study, we demonstrated that during inflammation, tumour-necrosis factor, IL-4R signalling and invariant Natural Killer T (iNKT) cells control the inducible formation of mesenteric FALCs[13]. However, the mechanisms controlling serous B-cell activation in FALCs and milky spots during immune challenge have not been fully defined. IL-33, a cytokine central to the activation of type 2 immune responses, has been shown to activate B1 B-cells to proliferate and secrete IgM *in vitro* and *in vivo* after intraperitoneal injection of recombinant IL-33 (ref. 20). Moreover, mesenteric FALCs are associated with the presence of ILC2s[14]. However, a direct *in vivo* link between type 2 inflammation, IL-33 release, ILC2s and serous B-cell responses has not been demonstrated.

As FALCs and milky spots are central to serous B-cell homeostasis and activation[13,15], here we investigate the physiological link between IL-33 signalling, FALCs and serous B-cell activation. We focus our study on the pleural cavity and the role of pericardial and mediastinal FALCs in pleural infection and airway inflammation. To understand the role of FALCs in pleural B-cell activation, we take advantage of the tissue tropism of the filarial nematode *Litomosoides sigmodontis* (Ls), a parasite that is restricted to the pleural cavity in its first stages of development[21]. In this study, we demonstrate that during Ls infection, mediastinal and pericardial FALCs support the activation of pleural B cells ensuring local secretion of IgM in the pleural space at the site of infection. Furthermore, we demonstrate that FALC B-cell activation during Ls infection is highly dependent on IL-33R signalling. Finally, using a model of lung allergic airway inflammation initiated by an extract of the fungus *Alternaria alternata* (Alt), we reveal a connection between lung inflammation and pleural FALC B-cell activation, in which IL-33 is crucial for rapid activation and localized secretion of IgM into the pleural space by FALC B cells. Importantly, we show that the stromal cells of pericardial and mediastinal FALCs produce IL-33 and that activation of FALC B1 cells by IL-33 is not direct, but requires secretion of IL-5 by IL-33 responsive cells.

## Results

**FALC B cells respond to parasite infection and secrete IgM.** Most studies to date that investigate B-cell responses to body cavity perturbations assessed the systemic production of antibodies within the serum[15,22–24]. After subcutaneous delivery of infective larvae, Ls migrate rapidly through the lymphatic system to the pleural cavity where the parasite resides. We reasoned that to be protective IgM would have to be produced locally by mediastinal and pericardial FALC B cells and secreted directly into the pleural space. We chose to assess resistant C57BL/6 mice at days 8–18 post infection, a time before immune mediated parasite killing but at which point an active immune response is occurring in the pleural cavity[25,26]. We compared the level of total and Ls-specific IgM secreted within the local environment (the lavage fluid of the pleural cavity), with the levels found in the serum and within lavage fluid of the peritoneal cavity, a site not related to the infection (Fig. 1a,b). Pleural lavage fluid had significantly increased levels of both antigen specific and total IgM 11 days post infection; whereas the peritoneal cavity showed no increase in IgM (Fig. 1a). Similarly, there was no infection-dependent increase in either total or antigen-specific IgM in the serum at this time point (Fig. 1b). These results indicated that during Ls infection, IgM production is initiated at the site of infection and that B cells present in the pleural environment were able to secrete IgM.

To determine if the FALCs of the pericardium and mediastinum (Fig. 1c) were involved in the pleural B-cell response against Ls, we analysed the effect of infection on the number, size and cell content of FALCs (Fig. 1d–f). Whole-mount immunofluorescence staining of mediastinal FALCs showed that both the size and number of CD45[+] clusters increased profoundly upon filarial infection by day 11 (Fig. 1d,e). In contrast, the number of FALCs of the mesenterium within the peritoneal cavity did not increase, indicating that the activation of FALCs is limited to the site of infection (Fig. 1e). We found that Ls infection induced a significant accumulation of immune cells both in pericardial FALCs and pleural exudate cells (PLEC), that continued to increase between day 0, day 11 and day 18 (Fig. 1f, first column). The differential expression of CD19, CD11b and CD5 was used to identify the innate like CD19[+]CD11b[+]CD5[+] B1a and CD19[+]CD11b[+]CD5[−] B1b-cell populations and the conventional CD19[+]CD11b[−]CD5[−] B2 cells (gated as shown in Supplementary Fig. 1). We found that all B-cell subsets were significantly increased in pericardial FALCs and PLEC upon infection (Fig. 1f, second, third and fourth columns).

To prove that FALCs were the source of antigen-specific IgM, equivalent numbers of peritoneal exudate cells, PLEC, digested pericardial FALC and lymph node cells from naive and Ls-infected animals were placed in overnight culture. Remarkably, all of the Ls-specific IgM was secreted by the pericardial FALC B cells, significantly more even than the cells of the draining mediastinal lymph node at this day 18 post-infection time point (Fig. 1g). These results indicated that during Ls infection, IgM production is initiated at the site of infection by FALC B cells.

To determine which of the B-cell subsets were producing IgM, pericardial and mediastinal FALC B1a, B1b and B2-cell populations isolated at day 18 post Ls infection were cell-sorted. After overnight culture, the amount of secreted Ls-antigen-specific and total IgM was determined by enzyme-linked immunosorbent

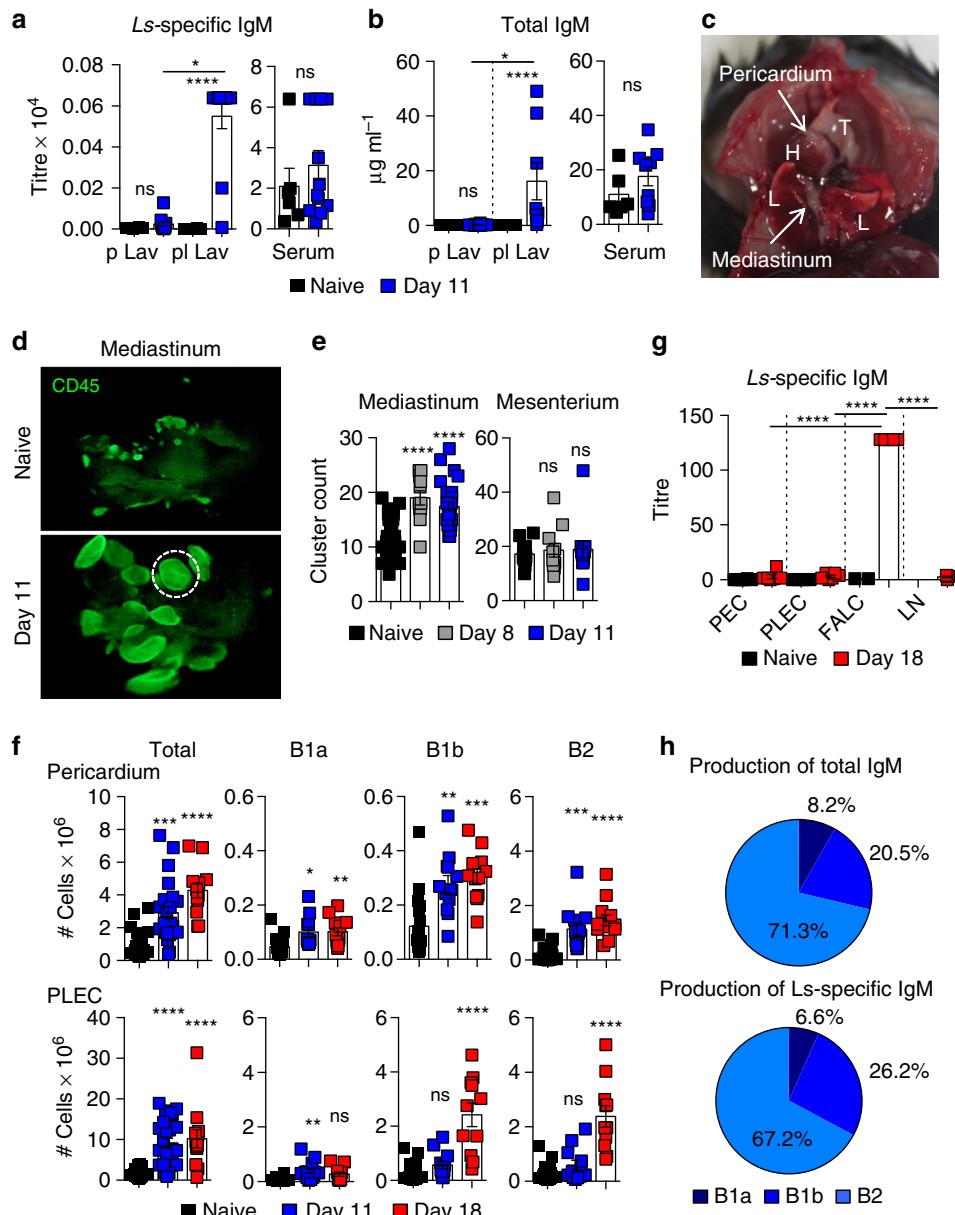

**Figure 1 | FALC B cells secrete parasite-specific IgM locally upon infection.** (**a**,**b**) Peritoneal lavage (p Lav), pleural lavage (pl Lav) and serum of naive and day 11 *Ls*-infected C57BL/6 mice were assessed for the presence of *Ls* antigen specific (**a**) and total IgM (**b**) by ELISA. Data shown combined from two representative experiments of >5 independent experiments n = 6 or 12 per group. (**c**) Location of the pericardium and mediastinum within the pleural cavity, thymus (T), Lungs (L) and Heart (H). (**d**) Representative whole-mount immunofluorescence staining of mediastina of naive and day 11 *Ls*-infected C57BL/6 mice showing CD45[+] clusters in green imaged with a fluorescence stereomicroscope. Original magnification × 4. (**e**) Quantification of the number of FALCs per mediastinum (left) and mesenterium (right) of naive, day 8 and day 11 *Ls*-infected C57BL/6 mice. Data combined from at least two independent experiments; symbols represent individual mice, n ≥ 11 per time point. (**f**) Flow-cytometric analysis of digested pericardial FALC cells and PLEC of naive and day 11 or 18 *Ls*-infected C57BL/6 mice. Numbers of each B-cell subset were quantified. Data combined from two independent experiments, symbols represent individual mice, n = 12 per time point. (**g**) Isolated peritoneal exudate cell (PEC), PLEC, FALC and lymph node (LN) cells of naive and day 18 *Ls*-infected C57BL/6 mice were cultured overnight and *Ls* antigen-specific IgM production quantified by ELISA. Data representative of two independent experiments; symbols represent individual mice n = 2, 4, 5 or 6 per group. (**h**) B1a, B1b and B2-cell populations isolated from pericardial and mediastinal FALCS from five individual day 18 *Ls*-infected C57BL/6 mice were cell-sorted as in (Supplementary Fig. 1) and total and *Ls*-specific IgM secretion was determined for each population by ELISA. The relative quantity of total and *Ls*-specific IgM produced by each cell type is shown. Data combined from two independent experiments. ns not significant, *P < 0.05, **P < 0.01, ***P < 0.001, ****P < 0.0001 (normally distributed data analysed by one-way ANOVA with Sidak multiple comparisons post test, non-normally distributed data analysed using the Kruksal–Wallis test with Dunn's multiple comparisons post test). Error bars represent mean with s.e.m. in all graphs.

assay (ELISA). While B1a, B1b and B2 cells had similar total IgM secretion capacity, B1b cells produced slightly more *Ls*-specific IgM than B1a cells (data not shown). When normalized to their relative cell numbers in FALCs, B2 cells were found to secrete the majority of the total and antigen-specific IgM (Fig. 1h). Thus the adipose depots of the pleural cavity support B1a, B1b and B2-cell antibody production, to enable provision of local IgM at the site of infection.

**FALC B2 cells differentiate into plasma cells in infected mice.** To further investigate the role of FALCs in B-cell activation, we performed whole-mount staining and confocal analysis of mediastinal FALCs from Day 11 *Ls*-infected mice. Staining of proliferative cells with the nuclear marker Ki67 revealed intense proliferation of IgM$^+$ and B220$^+$ B cells in FALCs of the infected mice (Fig. 2a confocal pictures and bar chart quantification of Ki67$^+$ pixels within FALCs). IgM staining revealed that a large proportion of B cells show accumulation of intra-cellular IgM, characteristic of IgM-secreting plasma cells (Fig. 2a). This was further confirmed by the induction of the plasma cell marker CD138 in FALCs of the infected mice (Fig. 2b).

As B2 cells produced the vast majority of *Ls*-specific IgM in FALCs, we focused our flow-cytometric analysis on the emergence of plasmablasts in the B2-cell subset from pericardial FALCs, PLEC and mediastinal LNs at day 11 post *Ls* infection. B2 plasmablasts were defined by the high expression of Ki67, accompanied by loss of membrane IgD (Fig. 2c, upper panel). Because CD138 is cleaved during collagenase digestion of FALCs, we identified plasma cells as SSC$^{high}$IgD$^-$ (Fig. 2c, lower panel). FALCs were the only compartment where active proliferation and B2-plasma cell differentiation occured (Fig. 2c–e). B2 cells of the

PLEC and mediastinal LNs in contrast, were not within active cell cycle (Fig. 2c,d). The large plasma cell population (30% of B2-cell subset) found in pericardial FALCs was not present in the draining LNs or the PLEC (Fig. 2c,e). Furthermore, we found a small population of GL7$^+$Ki67$^+$ germinal centre-like B2 cells within the pericardium at day 11 post infection (Fig. 2f). Taken together, these results provide evidence that pleural FALCs enable the formation of plasma cells-secreting antigen-specific IgM at the site of infection, outwith a classical secondary lymphoid organ.

**Parasite-specific IgM within pleural FALCs is dependent upon IL-33R.** IL-33, a cytokine central to the induction of type 2 responses has previously been shown to activate B1 cells to proliferate and secrete IgM *in vitro* and *in vivo* following intra-peritoneal injection of recombinant IL-33 (ref. 20). However, no direct *in vivo* link between type 2 immunity and IL-33-dependent activation of innate B-cell responses has been demonstrated so far. We thus assessed the role of the IL-33R in the activation of FALC B cells during *Ls* infection, which induces a type 2 immune response. Whole-mount immunofluorescence staining showed that the induction of B-cell proliferation in

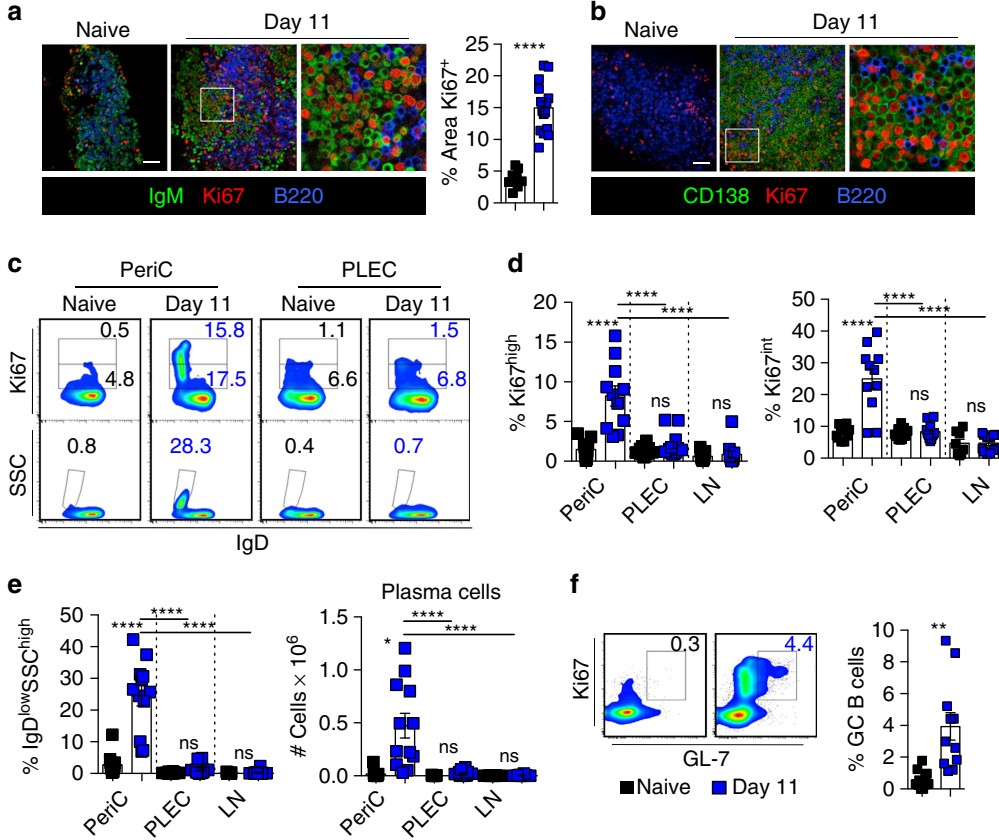

**Figure 2 | FALC B2 cells differentiate into plasma cells in parasite infected mice.** (**a**,**b**) Representative immunofluorescence staining of mediastinal FALCs of naive and day 11 *Ls*-infected C57BL/6 mice showing Ki67 (red), B220 (blue) and IgM (green in **a**) or CD138 (green in **b**) and quantification of the percentage area of Ki67 staining (**a**). Scale bar, 50 μm. (**c**–**e**) Flow-cytometric analysis of digested pericardial FALC cells and PLEC from naive and day 11 *Ls*-infected C57BL/6 mice showing Ki67$^{high}$ B2 cells (**c**, upper panel); and SSC$^{high}$IgD$^-$ plasma cells (**c**, lower panel). Quantification of % of Ki67$^{high}$ and Ki67$^{int}$ (**d**) and plasma cells (**e**, left graph) in B2 cells and plasma cell number (**e**, right graph) in pericardial FALCs, PLEC and lymph node (LN) are shown. (**f**) Flow-cytometric analysis of digested pericardial FALC cells from naive and day 11 *Ls*-infected C57BL/6 mice showing Ki67$^{high}$GL-7$^+$ germinal centre (GC) B cells and quantification of germinal centre B cells as a % of the total B2-cell population. Data in **a**-**f** are combined from two independent experiments, symbols in the graph in **a** represent individual clusters, data in **d**-**f** represent individual mice, $n = 12$ per time point. ns not significant, **$P < 0.01$, ****$P < 0.0001$ (normally distributed data analysed by one-way ANOVA with Sidak multiple comparisons post test, non-normally distributed data analysed using the Kruksal–Wallis test with Dunn's multiple comparisons post test). Error bars represent mean with s.e.m. in all graphs.

mediastinal FALCs of BALB/c mice was impaired in IL-33R-deficient BALB/c mice ($Il1rl1^{-/-}$) at day 11 post infection (Fig. 3a). Immunofluorescence staining indicated that mediastinal FALCs of naive $Il1rl1^{-/-}$ mice are smaller than their BALB/c counterparts, flow cytometric analysis of digested pericardial FALCs showed a trend for fewer cells in the mediastinal adipose but this did not reach significance (Fig. 3b). Next, we showed that in $Il1rl1^{-/-}$ mice, none of the pericardial FALC B-cell subsets increase in number following $Ls$ infection, unlike their wild type BALB/c counterparts (Fig. 3c). This suggested that IL-33R signalling is specifically needed for the accumulation of B cells in FALCs. Even though B2 cells failed to accumulate in pericardial FALCs of $Il1rl1^{-/-}$ mice, the differentiation of B2 cells into SSC$^{high}$IgD$^-$ plasma cells was not completly abrogated (Fig. 3d,e), suggesting that IL-33R is not, or only partially, involved in plasma cell differentiation. However, overall the total number of plasma cells found in FALCs was severely reduced (Fig. 3e). ELISA analysis confirmed the defect in pleural FALC B-cell activation with impaired IgM accumulation in the pleural lavage of $Il1rl1^{-/-}$ mice (Fig. 3f). Taken together, these results revealed that IL-33R signalling is essential for pleural B-cell antigen-specific IgM secretion during infection with the filarial nematode $Ls$.

**IL-33 is produced by FALC stromal cells.** Since B-cell activation in FALCs was so highly dependent on IL-33, we investigated whether pleural FALCs and milky spots could be a physiological source of IL-33. We compared the quantity of IL-33 released spontaneously by naive lung tissue (a known source of IL-33), the mediastinum, the omentum (peritoneal cavity adipose tissue containing milky spots) and the gonadal adipose tissue (GAT) (adipose tissue with few FALCs)[13], using a 1 h *in vitro* culture assay (Fig. 4a). Per gram of tissue, the mediastinum released 17 times more IL-33 than the GAT, and the omentum 7 times more IL-33 than the GAT. This suggested that the lymphoid clusters associated with adipose tissue of the visceral cavities are themselves a source of IL-33. The levels of IL-33 released by the mediastinum were comparable to the levels released by the lung. Since IL-33 expression has been previously reported in lymph node fibroblastic reticular cells[27], we performed whole-mount immunofluorescence staining and confocal analysis of mediastinal FALCs. The staining revealed that Gp38$^+$ stromal cells from mediastinal FALCs express high levels of nuclear IL-33 (Fig. 4b). Notably, we could also detect low levels of extra-nuclear IL-33 that colocalized with membranous Gp38 suggesting a basal release of IL-33 by FALC stromal cells. The same pattern of IL-33 expression was found in stromal cells of pericardial FALCs and omental milky spots (data not shown). We next addressed

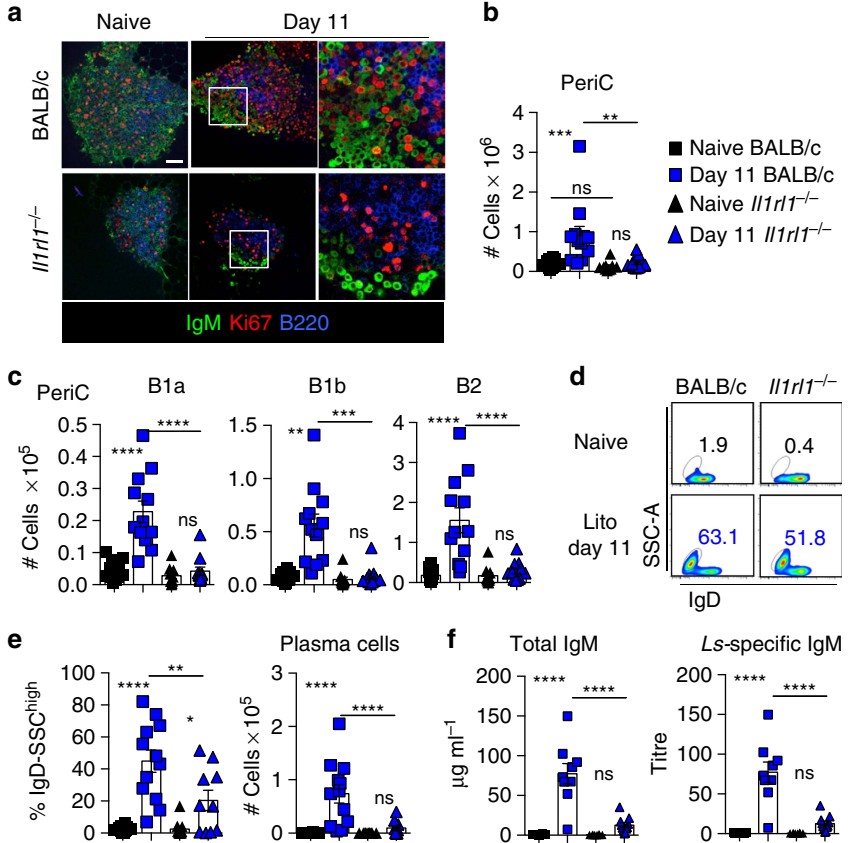

**Figure 3 | Parasite-specific IgM produced in FALCs is dependent upon IL-33R.** (**a**) Representative immunofluorescence staining of mediastinal FALCs from naive and day 11 $Ls$-infected BALB/c and $Il1rl1^{-/-}$ mice showing IgM (Green) Ki67 (Red) and B220 (blue). Scale bar, 50 μm. (**b,c**) Analysis of digested pericardial FALCs from naive and day 11 $Ls$-infected BALB/c and $Il1rl1^{-/-}$ mice showing total cell number and the number of B1a, B1b and B2 cells (gated as shown in Supplementary Fig. 1). (**d,e**) Flow cytometric analysis of digested pericardial FALCs from naive and day 11 $Ls$-infected BALB/c and $Il1rl1^{-/-}$ mice showing frequency of SSC$^{high}$IgD$^-$ B2 plasma cells (**d**) and quantification of frequency and number of SSC$^{high}$IgD$^-$ B2 plasma cells (**e**). (**f**) ELISA of total and $Ls$ antigen-specific IgM in the pleural lavage of naive and day 11 $Ls$-infected BALB/c and $Il1rl1^{-/-}$ mice. Data are combined from two independent experiments; symbols represent individual mice, $n \geq 8$ per time point. ns not significant, *$P < 0.05$, **$P < 0.01$, ***$P < 0.001$, ****$P < 0.0001$ (normally distributed data analysed by one-way ANOVA with Sidak multiple comparisons post test, non-normally distributed data analysed using the Kruksal–Wallis test with Dunn's multiple comparisons post test). Error bars represent mean with s.e.m. in all graphs.

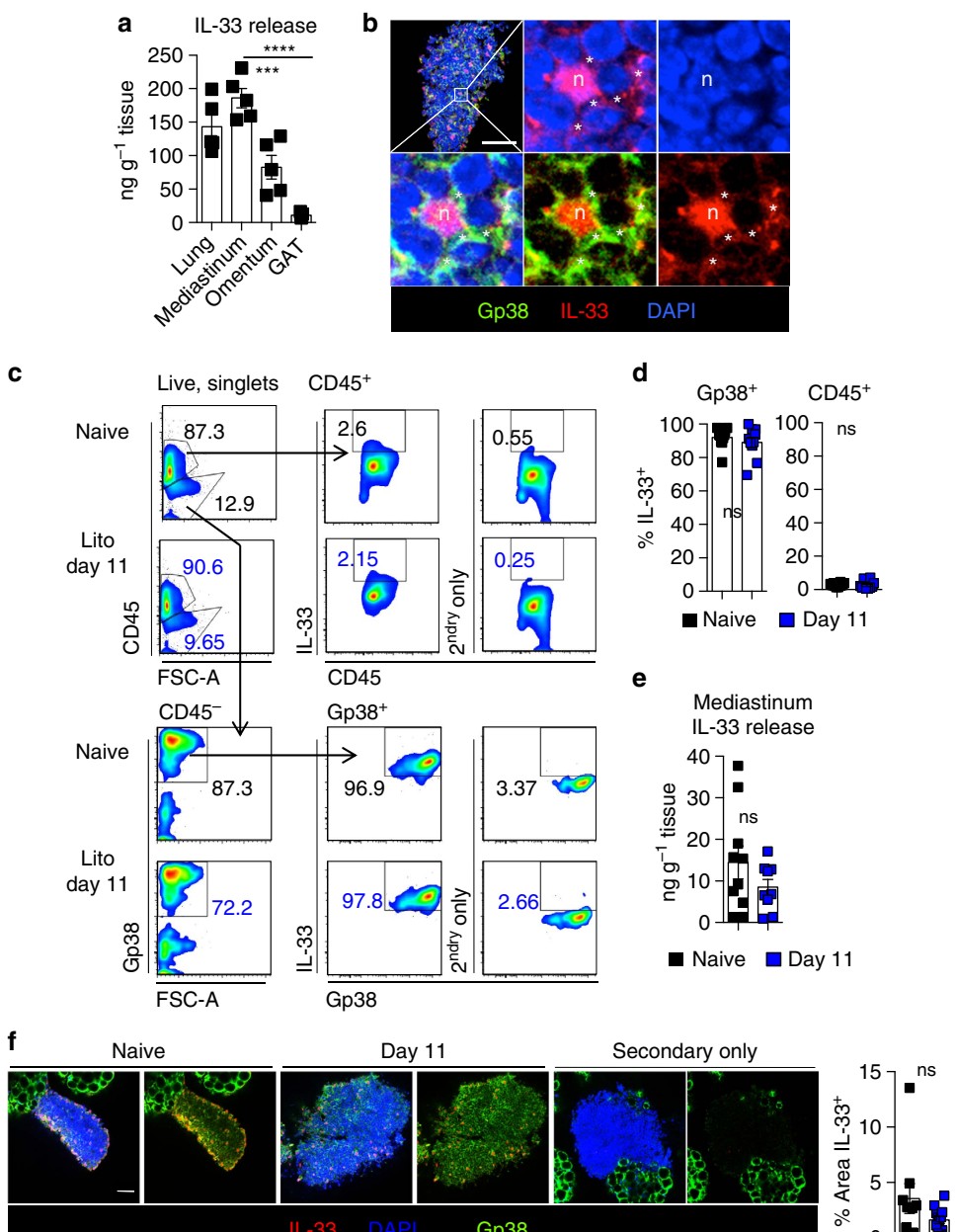

**Figure 4 | IL-33 is produced by FALC stromal cells.** (**a**) Pieces of lung, mediastinum, omentum or gonadal adipose tissue (GAT) were cultured for 1 h at 37 °C and spontaneous release of IL-33 in the supernatant was measured by ELISA. Data are representative of four independent experiments, symbols represent individual mice, n = 5. (**b**) Representative whole-mount immunofluorescence staining of mediastinal FALCs from BALB/c mice showing IL-33 (red), 4,6-diamidino-2-phenylindole (DAPI; blue) and Gp38 (green). Higher magnification shows expression of IL-33 in nucleus (n) and cytoplasmic/membranous area (*). Scale bar, 50 μm. (**c,d**) Flow cytometric analysis of digested pericardial FALCs from naive and day 11 *Ls*-infected C57BL/6 mice showing intracellular IL-33 expression by CD45⁻ Gp38⁺ stromal and CD45⁺ haematopoietic cells (**c**), and % of stromal cells and haematopoietic cells expressing IL-33 (**d**). (**e**) Naive and day 11 *Ls*-infected C57BL/6 mediastinum were cultured for 1 h at 37 °C and spontaneous release of IL-33 in the supernatant was measured by ELISA. (**f**) Representative whole-mount immunofluorescence staining of mediastinal FALCs from naive and day 11 *Ls*-infected C57BL/6 mice showing IL-33 (red), DAPI (blue) and Gp38 (green) and quantification of the % area of IL-33 staining. Scale bar, 50 μm. Data in (**c–f**) are pooled from two experiments, symbols in **d,e** represent individual mice, n = 8 or 10 per group, symbols in **f** represent individual clusters. ns not significant, ***P < 0.001, ****P < 0.0001 (normally distributed data analysed by one-way ANOVA with Sidak multiple comparisons post test, non-normally distributed data analysed using the Kruksal–Wallis test with Dunn's multiple comparisons post test). Error bars represent mean with s.e.m. in all graphs.

whether FALC stromal cells contained IL-33 during *Ls* infection (Fig. 4c). Flow cytometric analysis of digested pericardial FALCs confirmed that >95% of CD45⁻ GP38⁺ stromal cells from C57BL/6 mice expressed intracellular IL-33 (Fig. 4d) compared with only ~2% of CD45⁺ cells, when gated based on a secondary antibody only control (Fig. 4d). At day 11 following *Ls*

infection, no difference in the levels of IL-33 within stromal or haematopoietic cells was detected by flow cytometry (Fig. 4d), ELISA analysis of spontaneous IL-33 release during 1 h *in vitro* culture of the mediastinum (Fig. 4e) or whole-mount immunoflouresence staining as compared with naive controls (Fig. 4f).

**IL-33 controls pleural IgM secretion upon airway inflammation**. To address whether our findings with *Ls* were more broadly relevant, we investigated the activation of pleural FALCs in a distinct model in which involvement of the pleural cavity has never previously been addressed. Inhalation of inflammatory agents results in increased expression and secretion of IL-33 by lung epithelial cells[28] and intra-nasal (i.n.) instillation of the fungal allergen *Alt* is known to increase pulmonary IL-33 expression[28–31]. Surprisingly, i.n. delivery of *Alt* induced a noticeable release of IL-33 in the pleural lavage at 48 h (Fig. 5a) revealing continuity of the inflammatory IL-33 signal between the lungs and the pleural space.

We thus assessed whether the pleural IL-33 release induced by *Alt* instillation was associated with mediastinal B-cell activation. BALB/c and IL-33R-deficient ($Il1rl1^{-/-}$) mice were subjected to i.n. delivery of *Alt*, and B-cell activation in FALCs was assessed 48 h after instillation. Whole-mount immunofluorescence microscopy showed a marked induction of B-cell proliferation in mediastinal FALCs of BALB/c mice, which was impaired in IL-33R-deficient mice (Fig. 5b). We next quantified the precise effect of *Alt* instillation on B-cell proliferation in collagenase digested pericardial FALC cells and PLEC by flow cytometry using the nuclear marker Ki67. B cells in pericardial FALCs but not the PLEC showed high expression levels of Ki67 characteristic of cells actively in cell cycle 48 h after *Alt* instillation (Fig. 5c). Proliferation was more markedly increased in B1a and B1b cells (30% of Ki67^high B cells) than in B2 cells (3% of Ki67^high B2 cells) (Fig. 5c). Absence of Ki67^high B cells in PLEC demonstrated the importance of FALC to sustain pleural B-cell proliferation. Lack of IL-33R led to complete impairment in the induction of B1b and B2-cell proliferation and a fourfold reduction in the level of proliferation of FALC B1a cells in $Il1rl1^{-/-}$ mice (Fig. 5c). The marked proliferation of B1a and B1b-cell subsets in FALCs following *Alt* delivery in BALB/c mice was correlated with a significant increase in their number in pericardial FALCs but not in the PLEC (Fig. 5d). Absence of IL-33R abrogated the accumulation of B1a and B1b cells in FALCs after *Alt* instillation in $Il1rl1^{-/-}$ mice (Fig. 5d).

We then assessed the level of total IgM secreted within the pleural lavage, with the levels found within peritoneal lavage, a non-related site, and the serum. *Alt* instillation induced a significant increase in total IgM within the pleural lavage, but not in the peritoneal lavage or serum (Fig. 5e), demonstrating that the response induced is highly localized. Analysis of $Il1rl1^{-/-}$ mice showed that induction of IgM secretion upon *Alt* instillation in the pleural space is dependent on IL-33R. To confirm that FALC B cells were the source of the IgM produced in response to *Alt*, we placed the same number of wild-type pleural and mediastinal FALC elicited cells in culture overnight, and determined the production of total IgM within the supernatant by ELISA. In all, 200,000 FALC cells from *Alt* exposed animals secreted micrograms of IgM overnight (Fig. 5f). In contrast, PLEC were unable to secrete IgM (Fig. 5f) confirming in another setting that serous B cells do not secrete antibodies[32]. Taken together, these results show that FALCs of the pleural cavity respond to lung perturbation by supporting localized B1-cell proliferation and IgM production within the pleural cavity. Furthermore, IL-33R has a central role in the pleural B-cell response to acute lung inflammation controlling both FALC B1-cell proliferation and IgM secretion.

**FALC B cells require IL-5 for activation**. Peritoneal B cells were previously shown to express low levels of IL-33R and directly respond to IL-33 *in vitro*[20], suggesting that during acute lung inflammation the IL-33 released would act directly on pleural B cells. To test this, we labelled total PLEC from BALB/c and $Il1rl1^{-/-}$ donor mice with carboxyfluorescein succinimidyl ester (CFSE) and CellTrace Violet (CTV) respectively. Labelled PLEC were co-injected into the pleural space of a recipient BALB/c animal, *Alt* instilled 18 h later and the CFSE (wild type) and CTV ($Il1rl1^{-/-}$) labelled B-cell populations compared after 48 h (Fig. 6a). The injected PLEC were composed on average of 60% B cells of which 60% were B1 cells and 40% B2 cells. After transfer, between 2 and 4% of the PLEC were of donor origin. We could detect both CFSE and CTV-positive CD45$^+$CD19$^+$CD11b$^+$ B1 and CD45$^+$CD19$^+$CD11b$^-$ B2 cells within the PLEC and in the FALCs, indicating that IL-33R signalling was not necessary for the recruitment of pleural B cells into FALCs (Fig. 6b and not shown). Following *Alt* induction, the transferred B1 cells located within pericardial FALCs showed comparable dilution of CFSE and CTV, and the same increase in Ki67 expression (Fig. 6b), whereas the transferred cells found in the PLEC had not divided (not shown). This indicated that FALC resident $Il1rl1^{-/-}$ B1 cells had no impairment in the induction of proliferation and that intrinsic IL-33R signalling is not necessary for B1-cell proliferation upon *Alt* instillation.

It has been proposed that IL-33's action on B1 cells is largely mediated by IL-5 (ref. 20). We thus tested whether the action of IL-33 on FALC B1 cells was dependent on IL-5 in our model. At the time of *Alt* instillation, mice were injected intra-pleurally (i.pl.) with blocking anti-IL-5 antibody (Fig. 6c). Flow-cytometric analysis of FALC pericardial cells 48 h later showed that blocking of IL-5 completely abrogated B1a and B1b-cell proliferation compared with mice injected with control antibody (Fig. 6d). Confocal analysis confirmed that the level of proliferation induced by *Alt* in mediastinal FALCs was markedly reduced following i.pl. injection of blocking anti-IL-5 antibody (Fig. 6e,f). Blocking of IL-5 also led to impaired IgM secretion by B cells within the mediastinum (Fig. 6g). Taken together, these results indicate that IL-5 is critical for the activation of FALC B1 cells during *Alt* induced lung inflammation. As eosinophils are the other main target of IL-5[33,34], we wanted to assess the contribution of eosinophils to the induction of B-cell proliferation and IgM secretion. First, we analysed the impact of the delivery of anti-IL-5 antibody in the pleural cavity. As expected, we found a reduction in the number of eosinophils within the PLEC and a trend for reduced eosinophilia within the pericardial FALCs. However, this did not reach significance (Fig. 6h). To completely rule out that the effect we were seeing on B cells was dependent on eosinophils, we performed *Alt* experiments in Δ*dblGATA* mice that lack eosinophils. At 48 h following delivery of *Alt*, pericardial FALC B1a and B1b cells of Δ*dblGATA* mice were proliferating significantly more than their BALB/c counterparts (Fig. 6i), there was enhanced proliferation within the mediastinum as assessed by immunofluorescence staining (Fig. 6j) and there was no defect in the secretion of IgM within the pleural lavage (Fig. 6k). These data indicated that the induction of B-cell proliferation and IgM secretion was independent of eosinophils. However, since both B cells and eosinophils are dependent on IL-5, they may be in competition for its access. In the absence of eosinophils, B cells would have more IL-5 available, providing an explanation for the enhanced B-cell proliferation we found in Δ*dblGATA* mice.

**FALC ILC2s increase during pleural inflammation**. Finally, we determined the cellular origin of IL-5 in pericardial FALCs during *Ls* infection by analysing the intra-cellular levels of IL-5 within digested pericardium from C57BL/6 mice. We found here that Lineage$^-$CD90.2$^+$ innate lymphoid cells (ILCs), that represent 0.5–2% of total pericardial CD45$^+$ FALC cells, constitute the only reservoir of IL-5 producing cells within the pericardium (Fig. 7a–c). All other pericardial FALC cell populations assessed

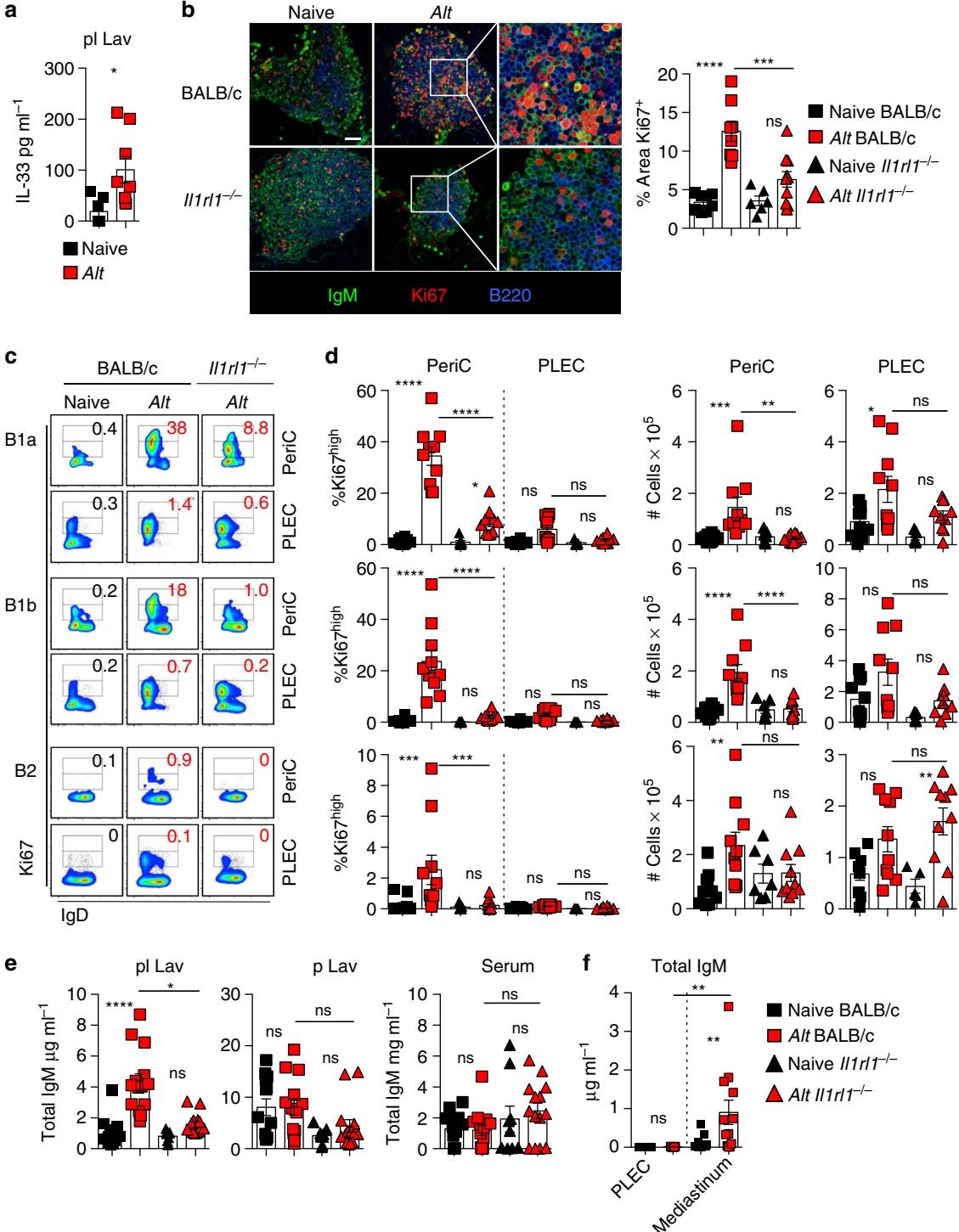

**Figure 5 | FALC activation during acute airway inflammation is IL-33 dependent. (a)** ELISA of IL-33 in the pleural lavage (pl Lav) of naive and *Alt*-treated C57BL/6 mice. Data are combined from two independent experiments, symbols represent individual mice, n = 6 or 9 per group. **(b)** Representative whole-mount immunofluorescence staining of mediastinal FALCs from naive and *Alt*-treated BALB/c and *Il1rl1*⁻/⁻ mice showing IgM (green), Ki67 (red) and B220 (blue) and quantification of the percentage area of Ki67 staining within individual clusters, scale bar, 50 µm. **(c,d)** Flow cytometric analysis showing proliferating Ki67^high and Ki67^int (**c**, left panel) and quantification of % Ki67^high (**c**, right panel) and number (**d**) of B1a (first row), B1b (second row) and B2 (third row) cells from digested pericardial FALC cells and PLEC from naive and *Alt*-treated BALB/c and *Il1rl1*⁻/⁻ mice. **(e)** ELISA of total IgM in the pleural lavage (pl Lav), peritoneal lavage (p Lav) and serum of naive and *Alt*-treated BALB/c and *Il1rl1*⁻/⁻ mice. Data in **c–e** are combined from two independent experiments, symbols represent individual mice, n = 8 or 10 per time point. **(f)** ELISA of total IgM in the supernatant of overnight culture of PLEC and digested FALC cells from naive and *Alt*-treated BALB/c mice. Data are combined from three independent experiments, symbols represent individual mice, n = 9 or 14 per time point. ns not significant, *P < 0.05, **P < 0.01, ***P < 0.001, ****P < 0.0001 (normally distributed data analysed by one-way ANOVA with Sidak multiple comparisons post test, non-normally distributed data analysed using the Kruksal–Wallis test with Dunn's multiple comparisons post test). Error bars represent mean with s.e.m. in all graphs.

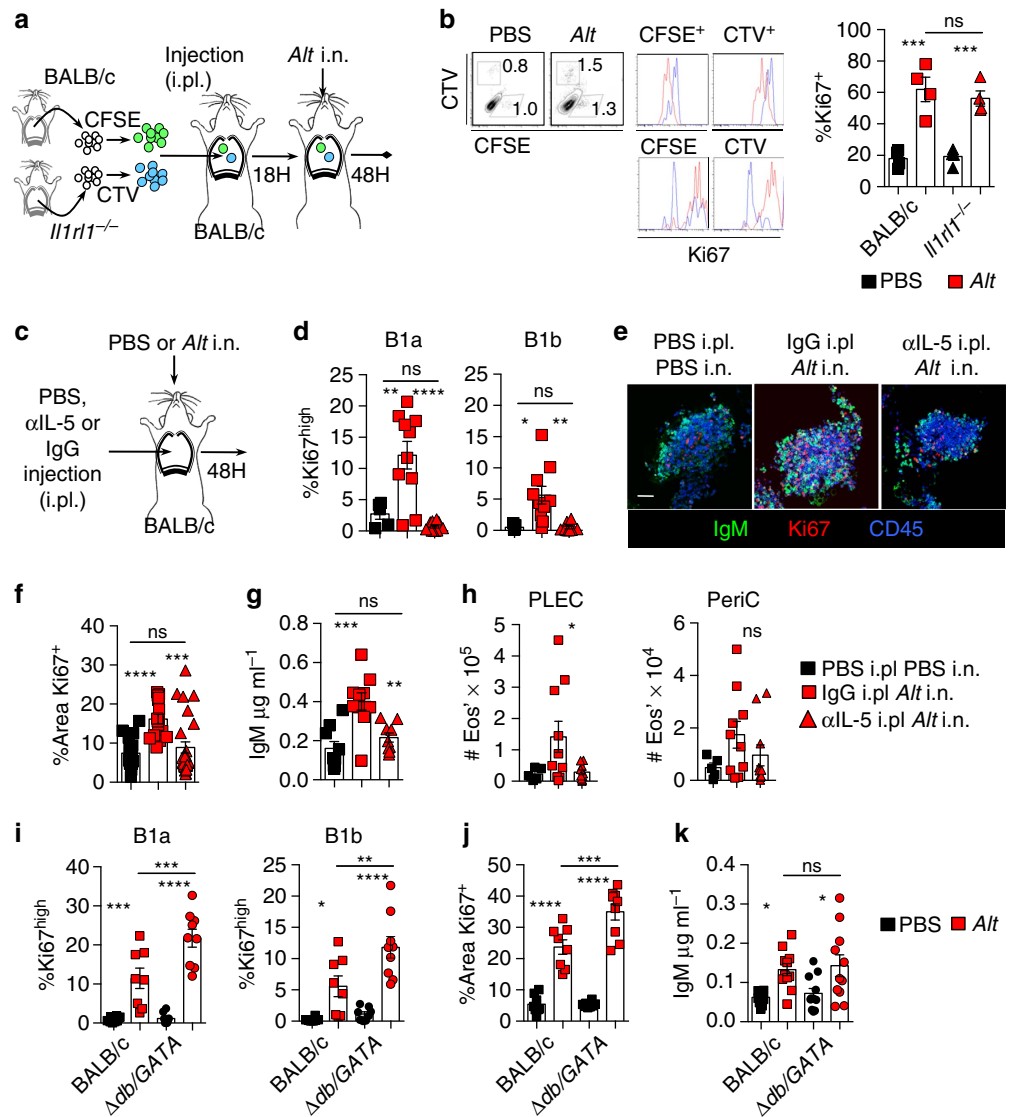

**Figure 6 | FALC B1-cell activation requires IL-5 but does not require eosinophils.** (**a**) Schematic of the i.pl. transfer experiment of CTV-labelled *Il1rl1*[−/−] and CFSE-labelled BALB/c control PLEC into BALB/c recipient mice before *Alt* treatment. (**b**) Flow cytometric analysis of digested pericardial and mediastinal FALCs showing gating of CFSE and CTV-labelled B1 cells (gated as CD45+CD19+CD11b+). CFSE and CTV intensities (upper histograms) and Ki67 expression (lower histogram) are shown for CFSE (left) and for CTV-labelled B1 cells (right). Quantification of %Ki67 expression within CFSE+ BALB/c and CTV+ *Il1rl1*[−/−] B1-cell populations. Data combined from two independent experiments, symbols represent individual recipient mice, *n* = 4. (**c-h**) Schematic of αIL-5 i.pl. blocking experiment with mice that received: intra-nasal (i.n.) and intra-pleural PBS, i.n. *Alt* and intra-pleural IgG control, i.n. *Alt* and intra-pleural anti-IL-5 blocking IgG (**c**). Flow cytometric analysis of digested pericardial FALCs, %Ki67high B1a and B1b cells in pericardial FALCs (**d**). Representative whole-mount immunofluorescence staining of mediastinal FALCs showing IgM (green), Ki67 (red) and CD45 (blue) (**e**). Scale bar, 50 μm. Quantification of % area of Ki67 staining (**f**). ELISA of total IgM secreted during 4 h of *in vitro* culture of whole mediastina (**g**). Quantification of eosinophil numbers within PLEC and digested pericardium (**h**). Data in **d–h** combined from 3 (**d–g**) and 2 (**h**) independent experiments, symbols represent individual mice, *n* = 13 or 15 (**d–g**) and *n* = 5 or 10 (**h**) per group. (**i–k**) BALB/c and Δ*dblGATA* mice received i.n. *Alt* for 48 h. Flow cytometric analysis of digested pericardial FALCs showing %Ki67high B1a and B1b cells (**i**). Quantification of the % area of Ki67 staining as assessed by whole-mount immunofluorescence staining of mediastinal FALCs (**j**). ELISA of total IgM within the pleural lavage (**k**). Data in **i–k** combined from two independent experiments, symbols represent individual mice, *n* = 8 or 9 per group, symbols in **j** represent individual clusters. ns not significant, *P < 0.05, **P < 0.01, ***P < 0.001, ****P < 0.0001 (normally distributed data analysed by one-way ANOVA with Sidak multiple comparisons post test, non-normally distributed data analysed using the Kruksal–Wallis test with Dunn's multiple comparisons post test). Error bars represent mean with s.e.m. in all graphs.

(CD45− Gp38+ stromal cells, CD19+MHC-II+ B cells, TCRβ+MHC-II− SSClo T cells, CD11b+ F4/80/Ly6c+ myeloid cells, Ly6G/SigF+SSChi MHC-II− Granulocytes) had no detectable intracellular IL-5 compared with the fluorescence minus one control (Fig. 7a). ILCs expressed significantly more IL-5 than all other cell populations assessed (Fig. 7b), however, there was no significant difference in the geometric mean fluorescence intensity of IL-5 expression when comparing naive and *Ls*

infection nor an increase in the percentage IL-5 expression within ILCs following infection (Fig. 7c). There was, however, a significant increase in the total number of ST2+ILC2s within the pericardium following *Ls* infection (Fig. 7d). IL-5+ ILCs were also present within digested pericardium from BALB/c mice and a trend towards an increase in the number of ST2+ILC2s was seen at 48 h following *Alt* instillation, however, this did not reach significance (Fig. 7e). Thus, our data indicate that increased

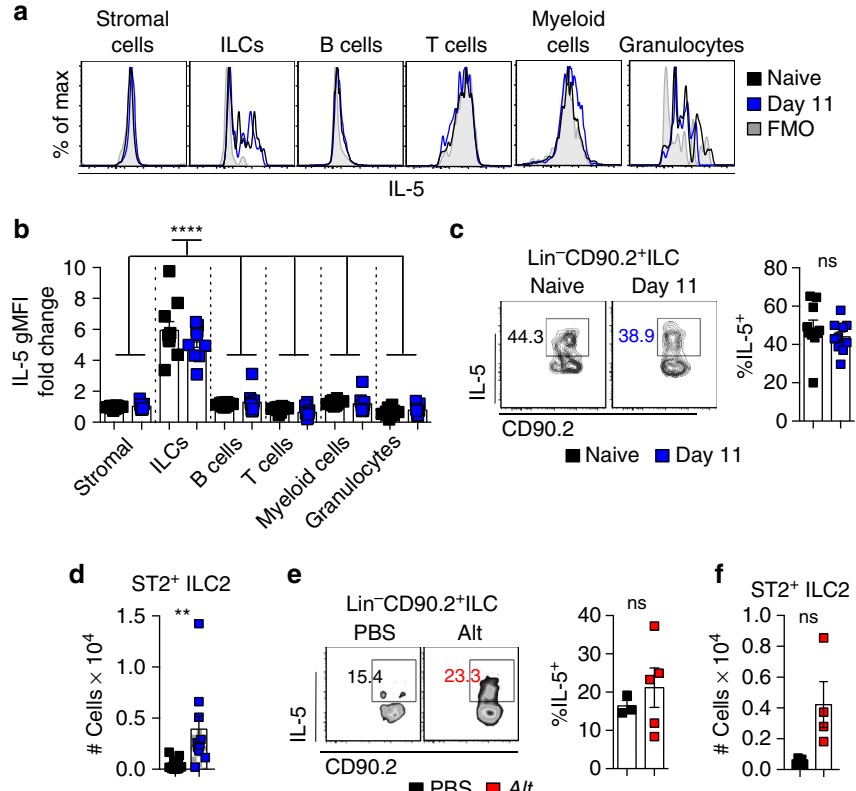

**Figure 7 | FALC ILC2s increase following induction of pleural inflammation.** (**a–c**) Flow-cytometric analysis of intracellular IL-5 in naive and day 11 *Ls*-infected C57BL/6 mice. Histogram analysis showing overlays of intracellular staining for IL-5 in naive mice (black histogram), *Ls*-infected mice (blue histogram) and fluorescence minus one (FMO) staining (grey histogram) within the indicated pericardial FALC cell populations (gated as described in main text) (**a**). Quantification of the geometric mean fluoresence intensity of IL-5 within these populations, expressed as fold change relative to the FMO control staining. ILC naive and infected samples were all **** increased over all other groups, there was no significant difference between naive and day 11 for any cell type (**b**). Dot plot analysis of intracellular IL-5 within pericardial FALC ILCs of naive and *Ls*-infected mice and quantification of the percentage of IL-5$^+$ ILCs (**c**). Data in **a**–**c** are pooled from two experiments, symbols represent individual mice, $n = 9$ or 10 per group. (**d**) Quantification of the number of Lineage$^-$CD90$^+$ST2$^+$ ILC2s within pericardial FALCs of naive and day 11 *Ls*-infected C57BL/6 mice. Data are pooled from four independent experiments, symbols represent individual mice, $n = 10$ or 14 per group. (**e,f**) Flow-cytometric analysis showing intracellular IL-5 within pericardial FALC ILCs following *Alt* treatment and quantification of the % of IL-5$^+$ ILCs (**e**). Quantification of the number of Lineage$^-$CD90.2$^+$ST2$^+$ ILC2s within pericardial FALCs in response to *Alt* treatment (**f**). Data in **e,f** are representative of two independent experiments, symbols represent individual mice, $n = 4$ or 5 per group. ns not significant, **$P < 0.01$, ****$P < 0.0001$ (normally distributed data analysed by one-way ANOVA with Sidak multiple comparisons post test, non-normally distributed data analysed using the Kruksal–Wallis test with Dunn's multiple comparisons post test). Error bars represent mean with s.e.m. in all graphs.

numbers of IL-5 producing ILC2s are the most likely source of IL-5 for FALC B cell activation following pleural inflammation induced by two distinct experimental models.

## Discussion

We show in two disparate models of infection and inflammation, that FALCs are sites of serous B-cell activation and plasma cell differentiation enabling localized production of IgM at the site of immune insult. We also demonstrate for the first time the biological significance of stromal-derived IL-33 for serous B-cell function. During pleural infection and acute lung inflammation, FALCs of the pleural cavity sustained pleural B-cell proliferation and local IgM secretion in an IL-33R-dependent manner. In response to acute lung inflammation FALC ILC2 rapidly produced IL-5, which was key to FALC B1-cell activation following lung inflammation. Therefore, we propose a model where early release of IL-33 from FALC stromal cells following pleural or lung perturbations leads to FALC ILC2 activation and IL-5 secretion. This then allows pleural B-cell recruitment to, and activation in mediastinal and pericardial

FALCs, resulting in active proliferation and increased production of local IgM (Fig. 8).

During the stage of infection where the filarial nematode *Ls* is confined to the pleural cavity, release of IgM in the serous cavity by FALC B cells represents a crucial source of protective antibodies, as serum IgM cannot diffuse into the body cavities. The critical importance of IgM for elimination from the peritoneal cavity of a closely related filarial larvae has been demonstrated using *sIgM$^{-/-}$* mice[24]. The protection conferred by IgM has been shown to be mediated through Fc receptor engagement and complement activation[35,36]. In addition, B1 cells are implicated in resistance to both *Ls*[22] and human filariasis[37]. However, it is not practical to remove FALCs from the pleural space during *Ls* infection, so we cannot directly address their role in protection. C57BL/6 μMT mice are not more susceptible to *Ls* primary infection[38] but these data are difficult to interpret because B cells are a major source of IL-10 (ref. 22), which is essential for susceptibility to *Ls*[39] and the development of female *Ls* adults is grossly impaired in the absence of IgM[23,40]. There is marked accumulation of M2 macrophages in the serous cavities of *Ls* mice[26] and thus it will be important to use more refined

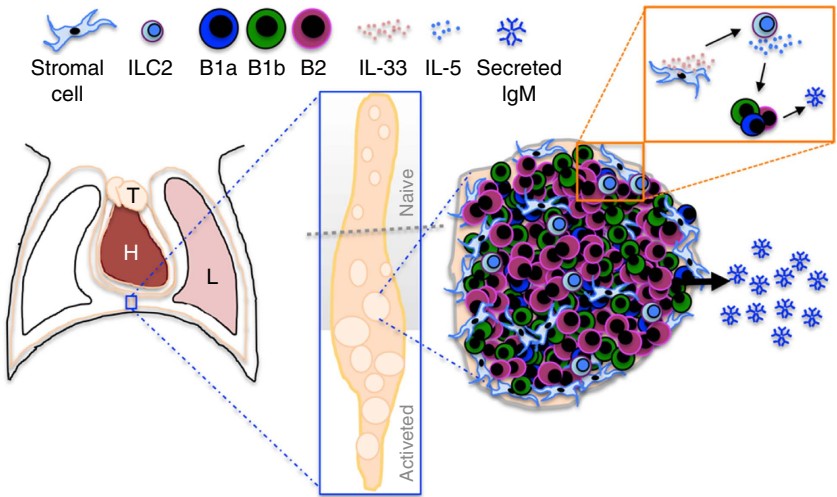

**Figure 8 | Graphical summary of data presented.** FALCs of the pleural cavity provide an IL-33 rich environment for rapid IgM producing B1-cell activation in response to lung inflammation or pleural infection. T = Thymus, H = Heart, L = Lung.

models to determine whether IgM recognition facilitates parasite killing by macrophages, as has been described in a related parasite model[41].

Transfer of *Il1rl1*-deficient pleural B cells into the pleural cavity of IL-33R competent mice showed that IL-33R on B cells is not necessary to induce pleural B-cell homing to FALCs, nor is it required for their proliferation (Fig. 3a,b). Furthermore, in contrast with previous reports[20], we could not convincingly detect ST2 expression on any pleural or FALC B-cell subsets neither in resting conditions nor upon *Ls* infection (data not shown). We think that ST2 expression and direct signalling of IL-33 in peritoneal B cells can only be achieved after several rounds of IL-33 injection, but that these doses were not physiologically achieved during *Alternaria* induced lung inflammation or *Ls* infection[20]. Although we cannot conclusively state that ILC2 are the critical source of IL-5 for pleural FALC B cells during IL-33-dependent responses, our data converges with Moro *et al.* [14] whose data show that mesenteric ILC2 secrete high amounts of IL-5 in response to IL-33. The importance of IL-5 in the context of pleural immune responses is supported by previous reports demonstrating that control of *Ls* infection is dependent on IL-5 (refs [42–44]). Our data further suggest that antibody mediated protection during filarial infection may well be controlled by IL-5.

The omentum has an important role in the priming of peritoneal B1 cells[10] which contribute to intestinal immunity by producing secretory IgA in the gut lamina propria[11,45,46]. Our data showing that lung immune inflammation results in pleural FALC B-cell activation suggests that pleural B cells and pleural FALCs may be for the lungs what peritoneal B cells and the omentum are for the gut. Recently, Weber *et al.* [47] showed that during microbial lung infection, pleural B1a cells relocate to the lung to produce IgM further supporting this concept. Natural IgM has proven to be particularly important for the control of airway infections of viral[48,49], fungal[50] and bacterial[47] origins but the mechanisms controlling IgM production and the sites of its secretion are still unclear. Harnessing the function of pleural FALCs to increase the protective effect of IgM may prove an interesting therapeutic target for the control of airway infection.

## Methods
**Animals and inflammation models.** Experiments were performed using male or female C57BL/6 (C57BL/6JOlaHsd from Harlan), BALB/c (BALB/cOlaHsd from Harlan), BALB/c *Il1rl1*$^{-/-}$ (ref. 51) and BALB/c Δ*dblGATA*[52] mice aged 8–12

weeks. Both lines have been backcrossed > 8 times onto the BALB/c background. All animals were bred and maintained under specific pathogen-free conditions at the University of Edinburgh Animal Facilities. Mice were infected sub-cutaneously with 25–30 *Ls* L3's or vehicle only or sensitized intra-nasally with 50 μg *Alt* extract (Greer) in PBS or PBS alone. All experiments were conducted under a license granted by the Home Office (UK) that was approved by the University of Edinburgh ethics committee. All individual experimental protocols are approved by the staff veterinarian before the start of the experiment.

**Cell isolation and culture.** Peritoneal exudate cell and PLEC were isolated by flushing the peritoneal and pleural cavities with RPMI 1640 (Sigma), respectively. Pericardium and mediastinum were enzymatically digested with 1 mg ml$^{-1}$ Collagenase D (Roche) for 35 min at 37 °C in RPMI 1640 containing 1% fetal bovine serum (Sigma). PLEC, lymph node cells or FALC cells, were cultured overnight and supernatant used for ELISA. Equivalent weights of lung, whole mediastinum, whole omentum and GAT were cultured for 1 h before determination of IL-33 in supernatant by ELISA. For analysis of intracellular IL-5 and IL-33, pericardium and PLEC cells were used directly *ex vivo* or rested overnight in RPMI 1640 (Sigma) containing 10% fetal bovine serum (Sigma), 50 U ml$^{-1}$ Penicillin (Sigma) 50 μg ml$^{-1}$ Streptomycin (Sigma), 2 mM L-glutamine (Sigma), cells were then stimulated for 4 h at 37 °C with 50 ng ml$^{-1}$ phorbol-myristate acetate (Sigma) and 1 μg ml$^{-1}$ Ionomycin (Sigma) including 1 × Brefeldin A (eBioscience) for the final 3 h of incubation.

**Flow cytometry.** Cells were surface stained, fixed and permeabilized as below. Cells were stained with LIVE/DEAD (Invitrogen), blocked with mouse serum and FcR-Block (clone 2.4G2, Biolegend), stained for cell surface markers (See Table 1 for list of antibodies used), and then fixed and permeabilized for intra-cellular staining with 1 × fixation-permeabilization buffer (eBioscience). Ki67 was detected using antibody clone B56 from BD Biosciences or antibody clone REA183 from Miltenyi Biotech, both of which allow the detection of Ki67$^{hi}$ cells. In Fig. 7: Lineage = CD19/TCRβ/Ly6G/Ly6C/F4/80/CD11c/FcεR1α/NK1.1/Ter119/CD5/CD11b/CD49b. Samples were acquired using a BD Fortessa and analysed with FlowJo software (Tree Star). Fluorescence activated cell sorting was performed using a BD FACSAria.

**Intra-pleural injections.** Total BALBc or *Il1rl1*$^{-/-}$ PLEC were labelled with 5 μg ml$^{-1}$ CFSE (Invitrogen) or 2 μM CellTrace Violet (Molecular probes), respectively, before injection of 200,000 labelled cells at a 1:1 ratio into the pleural cavity in 100 μl PBS. Fifty μg of *Alt* was delivered i.n. 18 h later. To block IL-5 within the pleural cavity, 30 μg of either purified functional grade anti-human/mouse IL-5 (eBioscience, clone TRFK5), 30 μg of Rat IgG1 (eBioscience, clone eBRG1) in 100 μl PBS (Sigma), or PBS (Sigma) alone was injected i.pl., immediately following injection 50 μl PBS or 50 μg of *Alternaria* extract (Greer) in 50 μl PBS (Sigma) was delivered i.n.

**Antibody and cytokine ELISAs.** IgM antibodies were detected using goat anti-mouse IgG/A/M (10100) from AbD Serotec diluted at 1 in 1000, followed by secondary goat anti-mouse IgM horseradish peroxidase (102005) from Southern Biotech diluted at 1 in 2000. Purified mouse IgM isotype control (553472) from BD Pharmingen was used as standard. For antigen-specific antibody detection, ELISA

**Table 1 | List of antibodies used in the flow cytometry and immunofluorescence stainings**

| Reagent | Clone | Conjugate | Source | Dilution |
|---|---|---|---|---|
| Armenian hamster anti-CD11c | N418 | PE | Biolegend | 1 in 200 |
| | | APC-Cy7 | | 1 in 200 |
| | | BV605 | | 1 in 100 |
| Armenian hamster anti-FcεRIα | Mar_1 | PE-Cy7 | Biolegend | 1 in 200 |
| Armenian hamster anti-TCRb | H57–597 | Pacific Blue | eBiosciences | 1 in 200 |
| Donkey anti-Goat | Polyclonal | AlexaFluor488 | Invitrogen | 1 in 300* |
| Donkey anti-mouse IgM | Polyclonal | Rhodamine Red | Jackson Laboratories | 1 in 500* |
| Goat anti-IL33 | Polyclonal | Unconjugated | R&D Systems | 1 in 25* |
| Golden Syrian hamster anti-Podoplanin (Gp38) | eBio1.8.1 | Biotin | eBiosciences | 1 in 100* |
| Mouse anti-CD3 | eBio500A2 | PE | eBiosciences | 1 in 200 |
| | | APC-Cy7 | | 1 in 200 |
| Mouse anti-Ki67 | B56 | FITC | BD Pharmingen | 1 in 100 |
| | REA183 | FITC | Miltenyi | 1 in 100 |
| | REA183 | PE | | 1 in 100 |
| | SolA15 | APC (IF) | eBiosciences | 1 in 100* |
| Mouse anti-NK1.1 | PK136 | PEVio770 | Miltenyi | 1 in 200 |
| Rat anti-I-A/I-E | M5/114.15.2 | AF700 | Biolegend | 1 in 800 |
| | | AF780 | eBiosciences | 1 in 800 |
| Rat anti-B220 | RA3-6B2 | eFluor450 | eBiosciences | 1 in 200* |
| Rat anti-CD11b | M1/70 | eFluor450 | eBiosciences | 1 in 200* |
| | | PerCP-Cy5.5 | | 1 in 1,200 |
| | | APC | | 1 in 800 |
| | | BV711 | Biolegend | 1 in 1,000 |
| | | PE Dazle − 594 | | 1 in 1,000 |
| Rat anti-CD138 | 281-2 | Biotin | Biolegend | 1 in 200* |
| Rat anti-CD19 | eBio1D3 | PE-Cy7 | eBiosciences | 1 in 200 |
| | 6D5 | PE | Biolegend | 1 in 200 |
| | 6D5 | BV421 | | 1 in 200 |
| Rat anti-CD4 | GK1.5 | eFluor660 | eBiosciences | 1 in 200 |
| | RM4–5 | BV650 | Biolegend | 1 in 200 |
| Rat anti-CD45.2 | 104 | FITC | eBiosciences | 1 in 100* |
| | | BV650 | Biolegend | 1 in 100 |
| | | PerCP | | 1 in 100 |
| Rat anti-CD49b | DX5 | APC-Cy7 | Biolegend | 1 in 200 |
| Rat anti-CD5 | 53-7.3 | APC-Cy7 | eBiosciences | 1 in 100 |
| | | PE-Cy7 | | 1 in 200 |
| Rat anti-CD90.2 | 30-H12 | Pacific Blue | Biolegend | 1 in 100 |
| | | PerCP | Miltenyi | 1 in 50 |
| Rat anti-F4/80 | BM8 | PE | eBiosciences | 1 in 200 |
| | | PE-Cy7 | | 1 in 200 |
| Rat anti-IgD | 11–26c | eFluor450 | eBiosciences | 1 in 200 |
| Rat anti-IL-5 | TRFK5 | PE | Biolegend | 1 in 100 |
| Rat anti-Ly-6C | HK1.4 | BV570 | Biolegend | 1 in 50 |
| | | PE-Cy7 | | 1 in 200 |
| Rat anti-Ly6G | 1A8 | BV421 | Biolegend | 1 in 200 |
| | | PE | BD Pharmingen | 1 in 200 |
| Rat anti-Siglec-F | E50–2440 | BV421 | BD Pharmingen | 1 in 200 |
| Rat anti-ST2 | DJ8 | Biotin | Mdbiosciences | 1 in 100 |
| | RMST2-2 | APC | eBiosciences | 1 in 100 |
| Rat anti-Ter119 | TER-119 | PE-Cy7 | eBiosciences | 1 in 200 |
| Streptavidin | | AlexaFluor555 | Invitrogen | 1 in 400* |
| | | PerCP | Biolegend | 1 in 200 |
| | | APC | | 1 in 200 |
| | | BV711 | | 1 in 200 |

APC, allophycocyanin; FITC, fluorescein isothiocyanate; IF, immunofluorescence; PE, phycoerythrin.
*Antibodies also used in immunofluorescence.

plates (Nunc) were coated with *Ls* Ag or *Alternaria* (Greer) at a concentration of $5\,\mu g\,ml^{-1}$ in carbonate buffer. IL-33 was detected using the IL-33 duoset ELISA (DY3626) from R&D systems following the manufacturers instructions. Plates were developed using two parts TMB reagent (KPL & Biolegend).

**Whole-mount immunofluorescence staining and confocal images.** Mediastinal adipose tissues were fixed in 10% neutral buffered formalin (NBF) for 1 h on ice and permeabilized in PBS 1% Triton for 30 min on ice prior staining with primary antibodies for 2 h at room temperature in PBS 0.5% BSA 0.5% Triton. After wash in PBS, tissues were stained with secondary antibodies for one hour at room temperature in PBS 0.5% BSA 0.5% Triton. Antibodies used are listed in Table 1. Confocal images were acquired using a Leica SP5 laser scanning confocal microscope. Image analysis was performed using ImageJ.

**Statistical analysis.** Power calculations showed that for our most commonly measured parameters (lymphoid cluster number and cell number) six mice per group provide sufficient power (90%) to detect at least a twofold difference between the groups, which we regard as an acceptable cutoff for identifying important biological effects. No randomization and no blinding was used for the animal experiments. Whenever possible, the investigator was partially blinded for assessing the outcome (cluster counts). All data were analysed using Prism 6 (Graphpad Prism, La Jolla, CA, USA).

**Data availability.** The authors declare that all relevant data supporting the finding of this study are available on request.

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

## Acknowledgements

We would like to thank E. Mohr, J. Caamano and members of the IIIR for their advice on the work, A. Fulton for maintaining the parasite life cycle, M. Waterfall for flow cytometry assistance, A. McKenzie and C. Lloyd for kind provision of mouse lines and the University of Edinburgh animal house staff and veterinarians for their husbandry. This work was supported by MRC UK grants to J.E.A. (MR/K01207X/1) and C.B. (MR/M011542/1).

## Author contributions

L.H.J.J. and C.B. designed the study, performed experiments, analysed data and wrote the manuscript. S.D. and M.S.M. performed experiments and analysed data, S.M.C. performed experiments, H.M. and R.M.M. provided biological resources and expert advice, J.E.A. oversaw the research and helped write the manuscript.

## Additional information

**Competing financial interests:** The authors declare no competing financial interests.

