## [Peer Review File · Nature Communications]

Reviewers' comments:

Reviewer #1

Expert in pulmonary immune responses

(Remarks to the Author):

Jackson-Jones et al. Fat associated lymphoid clusters control local IgM secretion during pleural infection and inflammation

This manuscript details experiments showing that the FALCs in the pericardium and mediastinum support IgM producing cells following infection with *Litomosoides sigmondontis* (Ls). Ls-specific IgM increased specifically in the pleural cavity and FALCs were significantly expanded in size and number after infection. Various B cell populations were modestly expanded, but the numbers of plasma cells were dramatically expanded in the pericardium following infection. IL-33 is important for control of Ls and in the absence of IL-33R, FALCs were decreased in size and the overall numbers of cells were reduced both before and after infection. IL-33R-deficiency impaired plasma cell formation and IgM secretion. The mediastinum was an important source of IL-33 (similar to lung). The authors also tested the role of FALCs and IL-33R in response to a fungal pathogen, *Alternaria* and observed similar results. Interestingly, although the B cell response was reduced in the absence of IL-33R, B cells did not respond directly to IL-33 and instead required IL-5.

This manuscript is very interesting and highlights the potential importance of an often-overlooked immunological compartment, the mediastinum and FALCs, in pulmonary and pleural immunity. However, the mechanistic studies linking IL-33R and IL-5 to the differentiation of B cells in this location are flawed in many ways.

1. The differentiation of PCs in IL-33R^{-/-} mice seems not as poor as the authors want us to believe. In figure 3C, there is a huge population of SSChiIgDlo cells in the IL-33R^{-/-} mice. The absolute numbers go down (probably because the total inflammatory infiltrate goes down), but their ability to differentiate seems fine.
2. It is interesting that the FALC size seems to go down in IL-33^{-/-} mice prior to infection (Fig 3a) and after infection. Is this observation supported by total cell numbers?
3. The division of cells into ki67^{hi} and KI67^{med} seems artificial and I have not seen this division before. I would gate all the KI67⁺ cells together. In this type of analysis, the flow plots (i.e in Fig 5c) would suggest much different conclusions. In fact, a huge proportion of B cells are responding in IL-33R^{-/-} mice. Again, I would conclude that B cell proliferation and plasma cell differentiation are probably OK in the IL-33R^{-/-} mice, but that the total numbers of cells is less.
4. The blocking antibody against IL-5 does not show that IL-5 acts on B cells - it may act on eosinophils. Have the authors performed B cell transfers with WT and IL-5R^{-/-} B cells?
5. One could make the argument that IL-33 is necessary for a type 2 inflammatory response, either from Th2 cells or ILC2 cells and that in the absence of IL-33, the whole response is impaired and fewer cells are recruited to the pleural cavity and mediastinum (data showing this are already published). Fewer eosinophils could mean less APRIL secretion and less support for plasma cells. IL-5 depletion may also reduce eosinophils and APRIL and lead to fewer PCs.
6. The authors make the case that the FALCs of the mediastinum and pericardium are the most important source of IgM production following Ls infection. This may be true, but any evidence is lacking. Is there a way to eliminate the FALCs or the B cells in the pleural cavity?
7. The authors focus on IgM, however, IgE is also an important Ig isotype in these infections. Is IgE produced in the FALCs?

Reviewer #2

Expert in FALC

(Remarks to the Author):

Jackson-Jones et al. examined the activation mechanisms of pleural B cells to produce antigen-specific IgM and the role of fat-associated lymphoid cluster (FALC) in response to filarial (*Litomosoides sigmodontis*) infection and fungus (*Alternaria*)-mediated lung inflammation and IgM production. The authors demonstrated in this paper that the activation, proliferation, plasmablast induction and IgM secretion take place in pericardial FALC but not in the serous of pleural cavity or lymph nodes. The authors further showed the importance of IL-33 produced by stromal cells of FALC using IL-33R-deficient mice but IL-33R on B cells is dispensable. In addition to IL-33, IL-5 was critical for the B cells activation and ILC2 seemed to be the source of IL-5 I response to IL-33.

This paper provides a new insight into our understanding of FALC for local IgM production. The experiments were performed in a logical manner and data are mostly convincing. There are, however, several points that need to be addressed.

1) In response to filarial infection, majority of total as well as antigen-specific IgM are produced by B2 cells (Fig. 1h). On the other hand, a lot more B1a and B1B cells are induced in *Alternaria* model (Fig. 5c, d) although antigen-specific response was not examined. In the latter model, B2 cells numbers were not affected by the lack of IL-33 receptor in PLEC. Is there any mechanistic difference between filarial infection model and *Alternaria* model?

2) It is intriguing why the authors did not examine the antigen-specific response in *Alternaria* model. In Fig. 5e and f, the authors should show *Alternaria*-specific IgM titers in addition to the concentration of total IgM.

3) In Fig. 6f, there seem to be two populations in anti-IL-5 treated group; half of population was not really affected by the anti-IL-5 treatment. Why?

4) ILC2 is able to produce IL-5 in response to PMA and ionomycin even in naïve mice. IL-33 produced by *Alternaria* treatment is expected to induce the proliferation of ILC2s. Therefore, the authors should show the absolute number of IL-5+ ILC2s in Fig. 6h.

5) Antigen-specific IgM is produced by FALC cells but not PLEC cells (Fig. 1g) but cell numbers in PLEC increased by *Alternaria* treatment (Fig. 5c, d), indicating that cells were mobilized to pleural cavity. It is intriguing that whereas adoptive transfer of PLEC cells resulted in the migration of B cells to FALC, antigen-specific B cells do not come out from the FALC to the pleural cavity. Where do PLEC cells come from? Are they from blood stream?

6) In Fig. 7, please add explanation of T, H, and L, which seem to be thymus, heart and lung.

7) On page 4 th line, "Type 2" Innate Lymphoid Cells should be "Group 2" Innate Lymphoid Cells.

Reviewer #3

Expert in IL-33, ILCs, lung immunology

(Remarks to the Author):

In the study titled, "Fat Associated Lymphoid Clusters control local IgM secretion during pleural

infection and lung inflammation" Jackson-Jones and colleagues investigate the role of B cells in Fat-associated Lymphoid Clusters (FALCs) in the pleural cavity. The authors describe an unexpected role for the mediastinal FALCs in innate immune responses in pleural nematode infection and lung inflammation, where IL-33, produced by adipose tissue stromal cells, activates group 2 innate lymphoid cells (ILC2) that produce IL-5 needed for pleural B1 cell activation and IgM production.

The current study is a continuum of an earlier one from the same authors demonstrating inflammation-induced expansion and activation of FALCs and FALC-associated B cells. However the notion of a large population of IgM+ B cells in the peritoneal and pleural cavities is not novel, their function has remained elusive. Jackson-Jones et al. have used two models of pleural inflammation characterizing B1 cell activation in FALCs. The most significant finding of the study is that pleural FALC sense and respond to lung inflammation, suggesting their FALC and FALC B cell involvement in various pathological conditions of the lung, such as infections, allergies or asthma.

The following points are raised:

- As the Ls parasite model is not widely used, would be great to have more details about infection, such as parasite burden on different days when samples were acquired. Also, references to relevant literature using the Ls model are missing (Babayán et al, Infection and Immunity 2003) or others.
- How relevant are B cells and IgM for Ls resistance? Some discussion of proof of this would be valuable.
- Figure 1 includes data acquired at several different timepoints, such as day 8, day 11 and day 18. What is the reasoning for that? What does it mean for the infection with Ls (parasite burden)?
- The mesenteric and pericardial adipose compartments are populated by ILC2s. ILC2s react to IL-33 and other epithelial factors induced by parasite infection by proliferation and production of IL-13 and IL-5. Is there an expansion/activation of ILC2 during Ls infection?
- Lung type 2 pneumocytes constitutively express IL-33 at high levels that is released rapidly upon infection, such as *Nippostrongylus brasiliensis* or lung inflammation (papain) model for example. The data in Figure 4 demonstrated that per g of tissue the mediastinum produces similar amount of IL-33 compared to the lung. However, in infection situation, there is so much more lung tissue that releases IL-33, making the relevance of IL-33 from FALC questionable.
- The data in Figure 4 to claim that IL-33 derives from stromal cell is weak. Why not use mesenteric FALCs as a comparison? It is known that GAT has little FALCs and is a very distant non-relevant tissue. Were negative/positive controls performed for the immunostaining? The data should be complemented by analyzing tissues from Ls infected animals.
- On Figure 5e it is evident that p Lavage provides folds more IgM than pl lavage even at steady state. This is not consistent with Fig.1, where total IgM levels at steady state in p lavage is close to 0. This should be discussed.
- What other cells produced IL-5 in the pericardium/lung? Do lung ILC2 produce IL-5?
- It is surprising that that ILC2s which are central in the story linking IL-33 and B cell-produced IgM, received so little attention in the analysis. Some inclusion of this analysis would improve the manuscript.
- The biological importance of the pathways described is not evident from the data presented. There is no connection to pathogen resistance or tissue protection in inflammation. Loss-of-function and gain-of-function experiments are missing for a high-impact journal!

Minor concerns:

- Fig 1c to supplementary figures. This is not part of relevant findings
- Figure 2a has no Ki67-positive B220+ cells, however this is claimed in the text
- Page 8, line 4 of 2nd paragraph word 'plasmablasts' misses an 's'. Same passage, line 10, 'pericardial' noncapitalize.
- The term 'FALC' is sometimes confusing in the text as it is hard to understand which

compartment is under discussion 'mediastinal FALC' or 'pericardial FALC' 'pleural FALC'. Page 10 talks about FALCs but figure 4 says "mediastinum". On figure 5 mediastinum has turned into 'mediastina". The is confusing to the reader.

Reviewer #1

Expert in pulmonary immune responses

(Remarks to the Author):

1. The differentiation of PCs in IL-33R^{-/-} mice seems not as poor as the authors want us to believe. In figure 3C, there is a huge population of SSC^{high}IgD⁻ cells in the IL-33R^{-/-} mice. The absolute numbers go down (probably because the total inflammatory infiltrate goes down), but their ability to differentiate seems fine.

We entirely agree with the analysis of the reviewer. This was already stated in the text: 'Even though B2 cells failed to accumulate in pericardial FALCs of *Il1rl1*^{-/-} mice, the differentiation of B2 cells into SSC^{high}IgD⁻ plasma cells was not completely abrogated (Fig. 3c and d), suggesting that IL-33R was only partially involved in plasma cell differentiation'. We now have modified this sentence as: 'Even though B2 cells failed to accumulate in pericardial FALCs of *Il1rl1*^{-/-} mice, the differentiation of B2 cells into SSC^{high}IgD⁻ plasma cells was not completely abrogated (Fig. 3c and d), suggesting that IL-33R was **not or** only partially involved in plasma cell differentiation'.

2. It is interesting that the FALC size seems to go down in IL-33^{-/-} mice prior to infection (Fig 3a) and after infection. Is this observation supported by total cell numbers?

Indeed in naïve animals, FALCs of IL-33R^{-/-} mice seemed smaller and there were less cells in the mediastinal adipose tissue of the IL-33R^{-/-} mice compared to BALB/c mice but this did not reach significance. A plot showing total cell numbers in pericardial adipose tissue has been added to Figure 3 and the following sentence has been added to the text: 'Immunofluorescence staining indicated that mediastinal FALCs of naïve *Il1rl1*^{-/-} mice were smaller than their BALB/c counterparts, flow cytometric analysis of digested pericardial FALCs showed a trend for fewer cells in the mediastinal adipose but this did not reach significance (Fig. 3b).'

3. The division of cells into ki67^{hi} and KI67^{med} seems artificial and I have not seen this division before. I would gate all the KI67⁺ cells together. In this type of analysis, the flow plots (i.e in Fig 5c) would suggest much different conclusions. In fact, a huge proportion of B cells are responding in IL-33R^{-/-} mice. Again, I would conclude that B cell proliferation and plasma cell differentiation are probably OK in the IL-33R^{-/-} mice, but that the total numbers of cells is less.

We realize that this important point needs clarification. Our group has assessed Ki67

co-staining with BrDu following a 3hr BrDu pulse while Phil Taylor's group has done similar experiments with pHH3 staining, which provides a definitive marker of proliferative events. We have both shown that cells in the M phase with a high level of Ki67 can be distinguished from cells in the S, G1 or G2 cells stained with intermediate levels of Ki67^{1 2}. This distinction is robust in multiple co-staining experiments. Intermediate levels of Ki67 are also retained in cells that have recently divided: eosinophils and monocytes that recently emigrated from the bone marrow, so total Ki67 staining needs to be used with caution. Critically, not all commercially available Ki67 antibodies allow this distinction to be made. The only antibodies we have found to do so are the B56 antibody clone from BD Biosciences and the REA183 antibody clone from Miltenyi biotech. This information has been further highlighted in the materials and methods.

4. The blocking antibody against IL-5 does not show that IL-5 acts on B cells - it may act on eosinophils. Have the authors performed B cell transfers with WT and IL-5R^{-/-} B cells?

IL-5 has two main cellular targets: B cells and eosinophils. We agree that transfer of IL-5R^{-/-} B cells would have proven that IL-5 directly acts on B cells but this mouse line was not available at our institution. To rule out that the effect we saw was dependent on eosinophils, we performed *Alternaria* experiments in eosinophil deficient mice and saw no difference in B cell proliferation in FALCs or level of IgM indicating that the induction of B cell proliferation and IgM secretion was independent of eosinophils. If anything we saw an increased proliferation of B cells in the absence of eosinophils that could be explained by the fact that in WT mice eosinophils and B cells compete for IL-5. In absence of eosinophils B cells have more IL-5 available and are more activated. These data have now been added to Figure 6 and the following paragraph added to the text:

“As eosinophils are the other main target of IL-5^{3, 4}, we wanted to assess the contribution of eosinophils to the induction of B cell proliferation and IgM secretion. First, we analysed the impact of the delivery of anti-IL-5 antibody in the pleural cavity. As expected, we found a reduction in the number of eosinophils within the pleural exudate and a trend for reduced eosinophilia within the pericardial FALCs. However, this did not reach significance (Fig. 6h). To completely rule out that the effect we were seeing on B cells was dependent on eosinophils we performed *Alt* experiments in $\Delta dbpGATA$ mice that lack eosinophils. At 48h following delivery of

Alt, pericardial FALC B1a and B1b cells of $\Delta dbfGATA$ mice were proliferating significantly more than their BALB/c counterparts (Fig. 6i), there was enhanced proliferation within the mediastinum as assessed by immunofluorescence staining (Fig. 6j) and there was no defect in the secretion of IgM within the pleural lavage (Fig. 6k). These data indicated that the induction of B cell proliferation and IgM secretion was independent of eosinophils. However, since both B cells and eosinophils are dependent on IL-5, they may be in competition for its access. In the absence of eosinophils, B cells would have more IL-5 available, providing an explanation for the enhanced B cell proliferation we found in $\Delta dbfGATA$ mice.”

5. One could make the argument that IL-33 is necessary for a type 2 inflammatory response, either from Th2 cells or ILC2 cells and that in the absence of IL-33, the whole response is impaired and fewer cells are recruited to the pleural cavity and mediastinum (data showing this are already published). Fewer eosinophils could mean less APRIL secretion and less support for plasma cells. IL-5 depletion may also reduce eosinophils and APRIL and lead to fewer PCs.

We concur that IL-5 does deplete eosinophils in the pleural cavity, however we do not see a significant reduction in the FALCs themselves. These data have been added to Figure 6 (h) and the text altered as above. In addition, we are aware that eosinophils are required for the maintenance of plasma cells in the bone marrow (Chu *et al* 2011) but we do not believe this will have a role at this early time point (48h) following pleural cavity activation

6. The authors make the case that the FALCs of the mediastinum and pericardium are the most important source of IgM production following *Ls* infection. This may be true, but any evidence is lacking. Is there a way to eliminate the FALCs or the B cells in the pleural cavity?

In this study we clearly showed that during *Ls* infection, the levels of IgM rose only in the pleural cavity and not the serum or the peritoneal cavity, which indicated that IgM was secreted only in the pleural cavity (Figure 1a-b). In this cavity, we demonstrated that only B cells in FALCs were competent to produce antibody as shown by our *in vitro* culture study (Figure 1g). We believe that this evidence though not entirely direct is strong. It is not currently possible to eliminate FALCs in the pleural cavity. Surgical removal of the pericardium is possible but total removal of FALC containing adipose is not. Furthermore, such surgery would disrupt the pleural membranes where the nematode

resides. We could use intra-pleural injection of anti-CD19 antibody to deplete B cells, however this wouldn't be a specific depletion of the cells in the FALCs but would target fluid phase B-cells as well, and thus we don't feel such experiments would add any new information to our manuscript.

7. The authors focus on IgM, however, IgE is also an important Ig isotype in these infections. Is IgE produced in the FALCs?

We typically have been unable to detect IgE within the substantial (2ml) volume of pleural lavage fluid that we routinely assessed and therefore had not pursued this isotype. However, because of the reviewers comment, we put the entire mediastinum in culture and tested for IgE. Indeed, we now have data showing that B-cells within the mediastinum can secrete IgE at day 11 following *Ls* infection when cultured for 4h *in vitro* (data to the right). Although this is interesting we have not included this in the revised text because the amounts are so small relative to IgM.

Reviewer #2
Expert in FALC
(Remarks to the Author):

Is there any mechanistic difference between filarial infection model and *Alternaria* model?

The two models we used are mechanistically very different. In the first model, the highly motile parasites are physically resident within the pleural cavity, whereas in the second model *Alternaria* is instilled via the nose, with a subsequent response occurring in the pleural cavity. Most importantly, we chose these two different models as they allow us to address very different FALC dynamics. The parasite model is assessed at 8-18 days following infection, by which point adaptive FALC B2 as well as innate B1a and B1b responses are occurring. The *Alternaria* model allows us to assess early innate cell responses, at a time point of 48 hours that is too early for both B2 and antigen-specific responses to have been initiated.

2) It is intriguing why the authors did not examine the antigen-specific response in *Alternaria* model. In Fig. 5e and f, the authors should show *Alternaria*-specific IgM titers in addition to the concentration of total IgM.

We do not find any antigen specific IgM at 48h following *Alternaria* instillation, we believe that this time point is too early for an antigen specific response to have been generated.

3) In Fig. 6f, there seem to be two populations in anti-IL-5 treated group; half of population was not really affected by the anti-IL-5 treatment. Why?

Yes, we noted this as well. The immunofluorescence analysis in Figure 6f includes the % area of Ki67 within whole 'FALCs' and not specifically B-cells within FALCs as assessed by flow-cytometry, thus non B-cell types within the clusters are the most likely explanation for this population that exhibits higher Ki67, and are unaffected by anti-IL-5 treatment.

4) ILC2 is able to produce IL-5 in response to PMA and ionomycin even in naïve mice. IL-33 produced by *Alternaria* treatment is expected to induce the proliferation of ILC2s. Therefore, the authors should show the absolute number of IL-5+ ILC2s in Fig. 6h.

We have removed the ILC data from Figure 6 and the absolute numbers of ILC2s have now been added to the new Figure 7 that also includes additional *Ls* ILC data. This paragraph has been added to the text:

FALC ILC2s increase following induction of pleural inflammation

Finally, we determined the cellular origin of IL-5 in pericardial FALCs during *Ls* infection by analyzing the intra-cellular levels of IL-5 within digested pericardium from C57BL/6 mice. We found here that Lineage⁻CD90.2⁺ ILCs, that represent 0.5-2% of total pericardial CD45⁺ FALC cells, constitute the only reservoir of IL-5 producing cells within the pericardium (Fig. 7a-c). All other pericardial FALC cell populations assessed (CD45⁻Gp38⁺ stromal cells, CD19⁺MHC-II⁺ B cells, TCRβ⁺MHC-II⁻SSC^{lo} T cells, CD11b⁺ F4/80/Ly6c⁺ myeloid cells, Ly6G/SigF⁺SSC^{hi} MHC-II⁻ Granulocytes) had no detectable intracellular IL-5 compared to the fluorescence minus one control (Fig. 7a). ILCs expressed significantly more IL-5 than all other cell populations assessed (Fig. 7b). However, there was no significant difference in the geometric mean fluorescence intensity (gMFI) of IL-5 expression when comparing naïve and *Ls* infection, nor an increase in the percentage IL-5 expression within ILCs following infection (Fig. 7c). There was however a significant increase in the total number of ST2⁺ILC2s within the pericardium following *Ls* infection (Fig. 7D). IL-5⁺ ILCs were also present within digested pericardium from BALB/c mice (Fig. 7e) and a trend toward an increase in the number of ST2⁺ILC2s was seen at 48h following *Alt* instillation, however this did not reach significance. Thus our data indicate that increased numbers of IL-5 producing ILC2s are the most likely source of IL-5 for FALC B cell activation following pleural inflammation induced by two distinct experimental models.”

5) Antigen-specific IgM is produced by FALC cells but not PLEC cells (Fig. 1g) but cell numbers in PLEC increased by *Alternaria* treatment (Fig. 5c, d), indicating that cells were mobilized to pleural cavity. It is intriguing that whereas adoptive transfer of PLEC cells resulted in the migration of B cells to FALC, antigen-specific B cells do not come out from the FALC to the pleural cavity. Where do PLEC cells come from? Are they from blood stream?

We believe that antigen-specific B cells are competent to migrate from the FALCS to the pleural cavity but when they do they are no longer able to produce antibodies since this is dependent on a close interaction with FALCs. FALC stromal cells provide the homeostatic chemokine Cxcl13 (previously shown in ⁵) and IL-33 creating a niche

bringing together B cells and ILC2 as a source of the IL-5 required to support antibody production. Other cells in the pleural influx are likely recruited from the blood stream or an expansion of resident cells. The dynamics of recruitment from the blood and proliferative expansion of PLEC cells under type 2 conditions has been described¹.

6) In Fig. 7, please add explanation of T, H, and L, which seem to be thymus, heart and lung.

We apologize for this oversight; the explanations for T, H and L have now been added to the figure legend for Fig 8 (previously Fig 7)

7) On page 4 th line, "Type 2" Innate Lymphoid Cells should be "Group 2" Innate Lymphoid Cells.

We have exchanged the word **Type** for the word **Group** in the text.

Reviewer #3

Expert in IL-33, ILCs, lung immunology

(Remarks to the Author):

- As the *Ls* parasite model is not widely used, would be great to have more details about infection, such as parasite burden on different days when samples were acquired. Also, references to relevant literature using the *Ls* model are missing (Babayan et al, *Infection and Immunity* 2003) or others.

Babayan *et al* 2003 has been added to the results section (Page 6, line 10). Details of the *Ls* model are also found in references 21, 22, 27, 29, 39, 40 and 41.

- How relevant are B cells and IgM for *Ls* resistance? Some discussion of proof of this would be valuable.

We have added a discussion of the evidence for B cells and IgM for *Ls* resistance to the discussion. In somewhat more detail:

IgM has been shown to be important for protection against infection with filarial worms in both experimental models⁶ and human studies⁷. Mishra *et al* found reduced B1 cell numbers and decreased IgM in microfilariae carriers as compared to endemic controls, suggesting that B1 cells are important for protection from filarial infection in the human. The BALB.Xid model in which B1 cells are deficient showed enhanced susceptibility to *Ls* infection (enhanced adult numbers and increased microfilariae) compared to BALB/c controls⁸. This data would support a functional role for FALC derived antibodies acting locally to limit L3 progression⁸.

Studies on full B cell KOs on the resistant C57BL/6 background are complex and do not provide straightforward answers. Although the evidence that B cells are needed for vaccine-mediated immunity is clear, in primary infection the C57BL/6 μ MT strain is not more susceptible⁹. However, B cells are a major source of IL-10 in this model⁸ and IL-10 is one of the most important susceptibility factors for *Ls*. It's absence will lead to parasite killing even in mice with no Th2 immunity¹⁰. Therefore in the absence of B cells, there will be less IL-10 and more effective worm killing, masking the absence of antibody. A further complication is the finding by Martin *et al* that the development of female filariae

are grossly impaired in the BALB/c μ MT strain¹¹. As these mice only lack IgM (unlike the C57BL/6 μ MT) it suggests that IgM is needed for proper worm development. We are excited to unravel this complicated system in future work but feel that it is currently outside the scope of this manuscript. We hope to acquire reagents and mouse lines in the future that will allow us to address the role of secreted IgM in resistance to this infection. We believe that the contribution of pleural FALCs to local IgM production remains an important novel finding.

The following paragraph has been added to the discussion:

In addition, B1 cells are implicated in resistance to both *Ls*²² and human filariasis²⁶. However, it is not practical to remove FALC from the pleural space during *Ls* infection, so we cannot directly address their role in protection. C57BL/6 μ MT mice are not more susceptible to *Ls* primary infection³⁸ but that data is difficult to interpret because B cells are a major source of IL-10²², which is essential for susceptibility to *Ls*³⁹ and the development of female *Ls* adults is grossly impaired in the absence of IgM^{27, 40}. There is marked accumulation of M2 macrophages in the serous cavities of *Ls* mice²⁹ and thus it will be important to use more refined models to determine whether IgM recognition facilitates parasite killing by macrophages, as has been described in a related parasite model⁴¹.

• Figure 1 includes data acquired at several different timepoints, such as day 8, day 11 and day 18. What is the reasoning for that? What does it mean for the infection with *Ls* (parasite burden)?

To be honest, Day 8, 11 and 18 were chosen because our initial hypothesis was that FALCs would be a major site of macrophage proliferation and these time points straddle the peak of proliferation. However, we made the novel discovery that FALCs were instead important site for B cell proliferation and antibody production. Nonetheless, we believe these are relevant time points as they allow us to assess the response at a stage of infection, when there is no significant difference in parasite burden between susceptible and resistant strains.

We have modified the following sentence in the text with the inclusion of additional references:

We chose to assess resistant C57BL/6 mice at days 8-18 post infection, a time prior to immune mediated parasite killing but at which point an active immune response is occurring in

the pleural cavity^{28, 29}

- The mesenteric and pericardial adipose compartments are populated by ILC2s. ILC2s react to IL-33 and other epithelial factors induced by parasite infection by proliferation and production of IL-13 and IL-5. Is there an expansion/activation of ILC2 during *Ls* infection?

We have added new data to Figure 7 showing that ILC2s are increased in the pericardial FALCs during *Ls* infection (Fig. 7a-d). The text has been modified as shown in response to reviewer 2, question 4.

Lung type 2 pneumocytes constitutively express IL-33 at high levels that is release rapidly upon infection, such as *Nippostrongylus brasiliensis* or lung inflammation (papain) model for example. The data in Figure 4 demonstrated that per g of tissue the mediastinum produces similar amount of IL-33 compared to the lung. However, in infection situation, there is so much more lung tissue that releases IL-33, making the relevance of IL-33 from FALC questionable.

We acknowledge that following major lung infection/inflammation total levels of IL-33 will be higher in this tissue, but we don't believe that this IL-33 will have a relevant role in the B cell processes we observe in the FALCs. Cytokines typically act locally in controlled cell-to-cell interactions. Even circulating cytokines are normally bound to carrier molecules and not necessarily functional. It would seem unlikely and potentially dangerous for IL-33 to act in a broadly systemic manner. It makes more biological sense for there to be local control with locally produced IL-33 activating local B cell responses. Indeed, our data highlight this point as we don't observe a full proliferative response in B-cells of the pleural cavity during either *Ls* or *alternaria*.

- The data in Figure 4 to claim that IL-33 derives from stromal cell is weak. Why not use mesenteric FALCs as a comparison? It is known that GAT has little FALCs and is a very distant non-relevant tissue. Were negative/positive controls performed for the immunostaining? The data should be complemented by analyzing tissues from *Ls* infected animals.

The mesenteric adipose has comparatively few FALCs in comparison to the omental adipose (see Benezech *et al* 2015), and thus omentum was used as a positive control. We used the GAT as a negative control site that lacks clusters, enabling us to extrapolate that IL-33 expression is linked to stromal cells within FALC containing tissues and not within adipose depots with a more dispersed immune network. In Figure

4 c-f we have now expanded our analysis of FALC stromal cell IL-33 production. This new data includes flow cytometric analysis of digested pericardium, including secondary only controls for IL-33 staining, whole mount immunofluorescence staining for IL-33 also including secondary only controls and ELISA analysis of mediastinum IL-33 release from both naïve and day 11 *Ls* infected C57BL/6 mice.

We have modified the text as follows:

‘We next addressed whether FALC stromal cells contained IL-33 during *Ls* infection (Fig. 4c). Flow cytometric analysis of digested pericardial FALCs confirmed that >95% of CD45⁺GP38⁺ stromal cells from C57BL/6 mice expressed intracellular IL-33 (Fig. 4d) compared to only ~2% of CD45⁺ cells, when gated based on a secondary antibody only control (Fig. 4d). At day 11 following *Ls* infection, no difference in the levels of IL-33 within stromal or haematopoietic cells was detected by flow cytometry (Fig. 4d), ELISA analysis of spontaneous IL-33 release during 1h *in vitro* culture of the mediastinum (Fig. 4e) or whole mount immunofluorescence staining as compared to naïve controls (Fig. 4f)’

- On Figure 5e it is evident that p Lavage provides folds more IgM than pl lavage even at steady state. This is not consistent with Fig.1, where total IgM levels at steady state in p lavage is close to 0. This should be discussed.

The explanation for the difference is that the data in Figure 1 are from C57BL/6 mice, while the data in Figure 5e are from BALB/c mice. A key point is that the peritoneal lavage reflects the serum IgM and naïve C57BL/6 animals have markedly lower levels of serum IgM (~10 µg) than naïve BALB/c animals (3 mg). The important finding is that in contrast to the pleural lavage, there is no increase in the amounts of IgM in the peritoneal lavage following *Ls* infection or *Alternaria* instillation, with the peritoneal lavage mirroring the serum response (though folds lower in the C57BL/6 mice).

- What other cells produced IL-5 in the pericardium/lung? Do lung ILC2 produce IL-5?

The only cells that we can show to be making IL-5 within the pericardium are ILCs, this data along with representative intracellular IL-5 staining of pericardial stromal cells, B-cells, T-cells, myeloid cells and granulocytes has been added to Figure 7. And the text

has been modified to state:

'We found here that Lineage⁻CD90.2⁺ ILCs, that represent 0.5-2% of total pericardial CD45⁺ FALC cells, constitute the only reservoir of IL-5 producing cells within the pericardium (Fig. 7a-c). All other pericardial FALC cell populations assessed (CD45-Gp38⁺ stromal cells, CD19⁺MHC-II⁺ B-cells, TCRβ⁺MHC-II⁻SSC^{lo} T-cells, CD11b⁺ F4/80/Ly6c⁺ myeloid cells, Ly6G/SigF⁺SSC^{hi} MHC-II⁻ Granulocytes) had no detectable intracellular IL-5 compared to the fluorescence minus one control (Fig. 7a). ILCs expressed significantly more IL-5 than all other cell populations assessed (Fig. 7b).'

- It is surprising that that ILC2s which are central in the story linking IL-33 and B cell-produced IgM, received so little attention in the analysis. Some inclusion of this analysis would improve the manuscript.

We have added additional ILC data from both *Ls* and *Alternaria* to revised Figure 7. Please see above for changes made to the text.

- The biological importance of the pathways described is not evident from the data presented. There is no connection to pathogen resistance or tissue protection in inflammation. Loss-of-function and gain-of-function experiments are missing for a high-impact journal!

We have spent considerable time considering how to directly address the role of the FALCs in resistance vs susceptibility to this parasite. The most obvious step, to specifically ablate the FALCs of the pleural space, is simply not possible. The use of B cell KOs is compromised as discussed above. However, we are excited to unravel this complicated system in future work and our next step will be to acquire mouse lines that will allow us to address the role of secreted IgM in resistance to this infection. However, as discussed above, the results may not provide immediate or straightforward answers. We believe our finding that pleural FALCs are a critical site for local IgM production remains an important novel finding with implications for a variety of conditions beyond this parasite model.

Minor concerns:

- Fig 1c to supplementary figures. This is not part of relevant findings

We believe that most readers will not be familiar with these tissues and that their visual

identification in figure 1c, will help with understanding. We can move this to the supplementary section if required.

- Figure 2a has no Ki67-positive B220+ cells, however this is claimed in the text.

Although the zoomed area of figure 2a which focuses on the strongly IgM positive region of the FALC has no B220+Ki67+ B cells, these cells are evident above and to the right of the white box in the middle panel of figure 2a.

- Page 8, line 4 of 2nd paragraph word 'plasmablasts' misses an 's'. Same passage, line 10, 'pericardial' noncapitalize.

These errors have been corrected in the text.

- The term 'FALC' is sometimes confusing in the text as it is hard to understand which compartment is under discussion 'mediastinal FALC' or 'pericardial FALC' 'pleural FALC'. Page 10 talks about FALCs but figure 4 says "mediastinum". On figure 5 mediastinum has turned into 'mediastina'. This is confusing to the reader.

We have altered Figure 5 so that it now reads **Mediastinum** rather than Mediastina and have modified the text in an attempt to be more consistent

REFERENCES

1. Jenkins, S.J. *et al.* Local macrophage proliferation, rather than recruitment from the blood, is a signature of TH2 inflammation. *Science* **332**, 1284-1288 (2011).
2. Davies, L.C. *et al.* Distinct bone marrow-derived and tissue-resident macrophage lineages proliferate at key stages during inflammation. *Nat Commun* **4**, 1886 (2013).
3. Yoshida, T. *et al.* Defective B-1 cell development and impaired immunity against *Angiostrongylus cantonensis* in IL-5R alpha-deficient mice. *Immunity* **4**, 483-494 (1996).
4. Kopf, M. *et al.* IL-5-deficient mice have a developmental defect in CD5+ B-1 cells and lack eosinophilia but have normal antibody and cytotoxic T cell responses. *Immunity* **4**, 15-24 (1996).
5. Benezech, C. *et al.* Inflammation-induced formation of fat-associated lymphoid clusters. *Nat Immunol* **16**, 819-828 (2015).
6. Rajan, B., Ramalingam, T. & Rajan, T.V. Critical role for IgM in host protection in experimental filarial infection. *J Immunol* **175**, 1827-1833 (2005).
7. Mishra, R. *et al.* Bancroftian filariasis: circulating B-1 cells decreased in microfilaria carriers and correlate with immunoglobulin M levels. *Parasite Immunol* **36**, 207-217 (2014).
8. Al-Qaoud, K.M., Fleischer, B. & Hoerauf, A. The Xid defect imparts susceptibility to experimental murine filariasis--association with a lack of antibody and IL-10 production by B cells in response to phosphorylcholine. *Int Immunol* **10**, 17-25 (1998).
9. Le Goff, L., Lamb, T.J., Graham, A.L., Harcus, Y. & Allen, J.E. IL-4 is required to prevent filarial nematode development in resistant but not susceptible strains of mice. *Int J Parasitol* **32**, 1277-1284 (2002).
10. Specht, S., Volkmann, L., Wynn, T. & Hoerauf, A. Interleukin-10 (IL-10) counterregulates IL-4-dependent effector mechanisms in Murine Filariasis. *Infect Immun* **72**, 6287-6293 (2004).
11. Martin, C. *et al.* B-cell deficiency suppresses vaccine-induced protection against murine filariasis but does not increase the recovery rate for primary infection. *Infect Immun* **69**, 7067-7073 (2001).

REVIEWERS' COMMENTS:

Reviewer #1 (Remarks to the Author):

For the most part the authors have appropriately addressed my previous comments. I still believe that proof of the the essential role of the mediastinal/pericardial FALCs is lacking (point 6), but as the authors point out, selectively eliminating the FALCs may be technically impossible. As a result, we have to rely on circumstantial evidence, which are reasonably compelling.

Reviewer #2 (Remarks to the Author):

Jackson-Jones et al. revised their paper with new experimental data. The authors promptly addressed most of the points raised by reviewers and the paper is much improved. There are still several minor points that need to be addressed.

1) The authors labeled PLEC with CFSE and CTV in the experiments shown in Fig. 6 and page 13. The cell composition of PLEC is unclear. The authors should provide the composition of PLEC such as X% of B1a, Y% of B1b, Z% of B2 etc. In addition, the authors should describe how they identified B1 cells in these experiments when they analyzed transferred cells.

2) The 4th line on page 7, PLEC should be spelt out here instead of those on the 12th line.

3) The 3rd line on page 10, (Fig. 3e) should be (Fig. 3e,f).

4) The 6th line from the bottom of page 15, (Fig. 7D) should be (Fig. 7d).

5) The 4th line from the bottom of page 15, (Fig. 7f) should be cited after "Alt instillation" or "did not reach significance".

6) At the end of the first paragraph of Discussion, (Fig. 7) should be (Fig. 8).

7) The 8th line of page 17, (Fig. 3A-D) should be (Fig. 3a-d).

8) In Materials and Methods, it should be described how many generations the authors backcrossed the *Il1r1^{-/-}* and Δ *dbpGATA* mice to BALC/c background.

9) The 5th line from the bottom of page 21, figure 7c should be figure 7b.

10) Although the authors mentioned in the legend for Fig. 2 that symbols in (a) represent individual clusters, there are no symbols but a square demonstrating the area for enlargement.

Reviewer #3 (Remarks to the Author):

Comments on the rebuttal letter from Jackson-Jones et al.

In general, the authors have significantly improved the manuscript by including novel data and modifying the text to facilitate an uninformed reader. I understand that gain- and loss-of-function experiments to address functional relevance of FALCs in physiology and infection are technically complicated due to unavailability of tools. However, I have some minor comments on modifications:

1) It is surprising that there is no change in IL-5-producing ILC2s, whereas, there is an increase in ILC2 numbers in the tissue upon Ls infection or alternaria. Is there a change when analyzing ILC2 proportions to CD45+ cells? The data on Fig. 7 is presented as change of FMO control staining that is very misleading at first glance. Raw MFI would be more informative. You are demonstrating fold change but apparently there is none when comparing naïve vs. infected condition. The rate of IL-5+ ILC2s is very high, higher than published for ILC2s in any other condition. Therefore, this data warrants for comparison with some tissue earlier published to include IL-5+ ILC2s, such as the colon.

2) On figure 4 you demonstrate the difference in IL-33 release from different tissues. Does IL-5 follow the same pattern? The ILC2s in FALC seem to be highly activated according to your data.

Reviewer #1 (Remarks to the Author):

For the most part the authors have appropriately addressed my previous comments. I still believe that proof of the the essential role of the mediastinal/pericardial FALCs is lacking (point 6), but as the authors point out, selectively eliminating the FALCs may be technically impossible. As a result, we have to rely on circumstantial evidence, which are reasonably compelling.

Reviewer #2 (Remarks to the Author):

Jackson-Jones et al. revised their paper with new experimental data. The authors promptly addressed most of the points raised by reviewers and the paper is much improved. There are still several minor points that need to be addressed.

1) The authors labeled PLEC with CFSE and CTV in the experiments shown in Fig. 6 and page 13. The cell composition of PLEC is unclear. The authors should provide the composition of PLEC such as X% of B1a, Y% of B1b, Z% of B2 etc. In addition, the authors should describe how they identified B1 cells in these experiments when they analyzed transferred cells.

The text and figure legend has been amended as shown below:

In the results section p13:

“To test this, we labeled total PLEC from BALB/c and *Il1r1*^{-/-} donor mice with CFSE and Cell Trace Violet (CTV) respectively, co-injected labeled PLEC into the pleural space of a recipient BALB/c animal, instilled *Alt* 18h later and compared the CFSE (WT) and CTV (*Il1r1*^{-/-}) labeled B cell populations after 48h (Fig. 6a). The injected PLEC were composed on average of 60% B cells of which 60% were B1 cells and 40% B2 cells. After transfer, between 2 and 4% of the PLEC were of donor origin. We could detect both CFSE and CTV positive CD45⁺CD19⁺CD11b⁺ B1 and CD45⁺CD19⁺CD11b⁻ B2 cells within the PLEC and in the FALCs, indicating that IL-33R signaling was not necessary for the recruitment of pleural B cells into FALCs (Fig. 6b and not shown).”

In the figure legend p28:

“Flow cytometric analysis of digested pericardial and mediastinal FALCs showing gating of CFSE and CTV labelled B1 cells (gated as CD45⁺CD19⁺CD11b⁺).”

2) The 4th line on page 7, PLEC should be spelt out here instead of those on the 12th line.

We have now corrected this in the text.

3) The 3rd line on page 10, (Fig. 3e) should be (Fig. 3e,f).

We apologize; we realize that the referencing of the figure was wrong in this paragraph. This has now been corrected this in the text.

4) The 6th line from the bottom of page 15, (Fig. 7D) should be (Fig. 7d).

We have now corrected this in the text.

5) The 4th line from the bottom of page 15, (Fig. 7f) should be cited after "Alt instillation" or "did not reach significance".

We have now corrected this in the text.

6) At the end of the first paragraph of Discussion, (Fig. 7) should be (Fig. 8).

We have now corrected this in the text.

7) The 8th line of page 17, (Fig. 3A-D) should be (Fig. 3a-d).

We have now corrected this in the text.

8) In Materials and Methods, it should be described how many generations the authors backcrossed the Il1r1^{-/-} and ΔdbpGATA mice to BALB/c background.

Both strains were backcrossed more than 8 times, this is now stated in the materials and methods.

9) The 5th line from the bottom of page 21, figure 7c should be figure 7b.

We have now corrected this in the text.

10) Although the authors mentioned in the legend for Fig. 2 that symbols in (a) represent individual clusters, there are no symbols but a square demonstrating the area for enlargement.

This has been now been corrected and clarified in the figure legend.

Reviewer #3 (Remarks to the Author):

In general, the authors have significantly improved the manuscript by including novel data and modifying the text to facilitate an uninformed reader. I understand that gain- and loss-of-function experiments to address functional relevance of FALCs in physiology and infection are technically complicated due to unavailability of tools. However, I have some minor comments on modifications:

1) It is surprising that there is no change in IL-5-producing ILC2s, whereas, there is an increase in ILC2 numbers in the tissue upon Ls infection or alternaria. Is there a change when analyzing ILC2 proportions to CD45+ cells? The data on Fig. 7 is presented as change of FMO control staining that is very misleading at first glance. Raw MFI would be more informative. You are demonstrating fold change but apparently there is none when comparing naïve vs. infected condition. The rate of IL-5+ ILC2s is very high, higher than published for ILC2s in any other condition. Therefore, this data warrants for comparison with some tissue earlier published to include IL-5+ ILC2s, such as the colon.

Even though the number of ILC2 in FALCs increase during Ls infection and upon Alternaria induced lung inflammation, the proportion of ILC2 in the CD45⁺ population remains stable (not shown). This is not surprising considering the massive expansion of B cells in FALCs during these immune challenges. Raw MFI analysis showed clearly that in FALCs only ILC2 are positively stained for intra-cellular IL-5 after *ex vivo* stimulation when

compared to FMO control staining. See Figure below. Fold change to FMO control staining is a common way of normalizing gMFI data from cell populations with different auto-fluorescence levels. To keep our figure easy to read, we would like to present the data with the normalized gMFI.

ILC2s are the main IL-5 producing cells in FALCs: Flow cytometric analysis of intracellular IL-5 in naïve and day 11 *Ls* infected C57BL/6 mice. Quantification of the Geometric Mean Fluorescence Intensity of intracellular IL-5 and FMO control staining within the indicated cell populations, from naïve and day 11 *Ls* infected C57BL/6 mice. Only ILC2 from naïve & infected samples showed an increase gMFI of IL-5 compared to FMO control staining.

Molofsky et al. showed using IL-5 reporter mice that most adipose tissue ILC2 spontaneously produced IL-5. Upon *ex vivo* stimulation with PMA+ionomycin or IL-33, naïve ILC2 secreted high amounts of IL-5 (Molofsky et al., J. Exp. Med., 2013). Moro et al., also showed that naive ILC2 purified from mesenteric adipose tissue, which also contains FALCs, were very potent producer of IL-5 in response to PMA + ionomycin or IL-33 (Moro et al., Nature, 2010). Our data thus confirm that ILC2 are very potent producer of IL-5. *Ex vivo* stimulation of ILC2 by PMA and Ionomycin is likely to override the intrinsic IL-5 production induced by *Ls* infection or *Alternaria* inflammation explaining why we did not detect any differences between naïve and challenged ILC2.

2) On figure 4 you demonstrate the difference in IL-33 release from different tissues. Does IL-5 follow the same pattern? The ILC2s in FALC seem to be highly activated according to your data.

As discussed in 1), Molofsky et al. demonstrated with reporter mice that gonadal adipose tissue ILC2s were prone to produce IL-5. We found that the levels of IL-33 produced per gram of tissue by the gonadal adipose was low compared to the omentum or mediastinum, indicating that the high level of IL-5 production in naïve mice we observed in pericardial and mediastinal FALCs was not directly correlated to the amount of IL-33 produced per g of tissue. However, we think that inside FALCs, IL-33 producing stromal cells provide a IL-33 rich environment enabling ILC2 to produce IL-5 and allowing maintenance of IL-5 dependent B1 cells. The existence and the nature of a niche for an IL-33 dependent activation of ILC2 in gonadal adipose tissue remain to be investigated.